# The non-muscle actinopathy-associated mutation E334Q in cytoskeletal γ-actin perturbs interaction of actin filaments with myosin and ADF/cofilin family proteins

**Johannes N Greve[1], Anja Marquardt[1], Robin Heiringhoff[1], Theresia Reindl[1†], Claudia Thiel[1], Nataliya Di Donato[2], Manuel H Taft[1], Dietmar J Manstein[1,3,4]***

[1]Institute for Biophysical Chemistry, Hannover Medical School, Fritz Hartmann Centre for Medical, Hannover, Germany; [2]Department of Human Genetics, Hannover Medical School, Hannover, Germany; [3]Division for Structural Biochemistry, Hannover Medical School, Hannover, Germany; [4]RESiST, Cluster of Excellence 2155, Hannover Medical School, Hannover, Germany

*For correspondence:
manstein.dietmar@mh-hannover.
de

Present address: †Department of Microbiology and Immunology, Stanford University School of Medicine, Stanford, United States

Competing interest: The authors declare that no competing interests exist.

**Abstract** Various heterozygous cytoskeletal γ-actin mutations have been shown to cause Baraitser–Winter cerebrofrontofacial syndrome, non-syndromic hearing loss, or isolated eye coloboma. Here, we report the biochemical characterization of human cytoskeletal γ-actin carrying mutation E334Q, a mutation that leads to a hitherto unspecified non-muscle actinopathy. Following expression, purification, and removal of linker and thymosin β4 tag sequences, the p.E334Q monomers show normal integration into linear and branched actin filaments. The mutation does not affect thermal stability, actin filament nucleation, elongation, and turnover. Model building and normal mode analysis predict significant differences in the interaction of p.E334Q filaments with myosin motors and members of the ADF/cofilin family of actin-binding proteins. Assays probing the interactions of p.E334Q filaments with human class 2 and class 5 myosin motor constructs show significant reductions in sliding velocity and actin affinity. E334Q differentially affects cofilin-mediated actin dynamics by increasing the rate of cofilin-mediated de novo nucleation of actin filaments and decreasing the efficiency of cofilin-mediated filament severing. Thus, it is likely that p.E334Q-mediated changes in myosin motor activity, as well as filament turnover, contribute to the observed disease phenotype.

## eLife assessment

This study presents a **valuable** characterization of the biochemical consequences of a disease-associated point mutation in a nonmuscle actin. The study uses well-characterized in vitro assays to explore function. The data are **convincing** and should be helpful to others.

## Introduction

Cytoskeletal β- and γ-actin are the basic building blocks of the actin cytoskeleton in human non-muscle cells. The spatio-temporal regulation of these two actin isoforms, mediated by distinct actin-binding proteins (ABPs), drives the establishment and maintenance of specialized cytoskeletal compartments that are defined by their network architecture and the presence of specific

filamentous actin-associated proteins (*Pollard, 2016*; *Plastino and Blanchoin, 2018*). The regulated ATP-dependent turnover of these actin networks as well as the coordinated interplay of actin with network-specific ABPs contributes to essential cellular processes like directed migration, adhesion, division, and mechanotransduction (*Ponti et al., 2004*; *Abercrombie et al., 1970*; *Abercrombie et al., 1971*; *Ridley et al., 2003*; *Pollard et al., 2002*; *Lauffenburger and Horwitz, 1996*).

β- and γ-actin differ by only four amino acids in the N-terminal region of the molecule but show isoform-specific spatial organization in studies performed with monoclonal antibodies. In motile and stationary cells, β-actin is found in contractile stress fibers and the cell periphery (*Dugina et al., 2009*; *Latham et al., 2013*; *Dugina et al., 2015*). In contrast, γ-actin localization is less static. During cell movement, γ-actin is recruited to the leading edge of the cell, resulting in an enrichment of the isoform in the lamellipodium together with β-actin (*Dugina et al., 2009*). In non-moving cell states, γ-actin is co-localized with β-actin in stress fibers (*Dugina et al., 2009*). In order to reconcile the highly similar protein sequences with the varying localization, several hypotheses on isoform regulation were put forward. These include the preferential interaction of ABPs with a particular isoform depending on the minor differences in the protein sequence, post-translational modifications as a means of isoform-specific regulation, and differences in the translation speed of *ACTB* and *ACTG1*, the genes encoding cytoskeletal β-actin and γ-actin (*Zhang et al., 2010*; *Vedula et al., 2017*; *Pavlyk et al., 2018*; *Müller et al., 2013*).

Diseases associated with heterozygous pathogenic variants in *ACTB* and *ACTG1* are typically referred to as non-muscle actinopathies. These include a spectrum of rare syndromes that manifest themselves as more or less pronounced developmental disorders in patients (*Verloes et al., 2015*; *Di Donato et al., 2014*; *Latham et al., 2018*; *Rivière et al., 2012*; *Baraitser and Winter, 1988*; *Cuvertino et al., 2017*). Baraitser–Winter cerebrontofacial syndrome (BWCFF, *Baraitser and Winter, 1988*) is a condition that affects the development of multiple organs, but particularly results in facial anomalies and brain malformations. BWCFF occurs due to various missense mutations in *ACTB* or *ACTG1* (*Verloes et al., 2015*; *Di Donato et al., 2014*; *Rivière et al., 2012*; *Di Donato et al., 2016*). BWCFF patients show a recognizable facial gestalt, with variants in *ACTB* generally resulting in a stronger facial phenotype than mutations in *ACTG1* (*Di Donato et al., 2014*). BWCFF patients regularly suffer from cortical malformations that lead to severe developmental and mental impairments. In addition, symptoms such as hearing loss, dystonia, hypotonia, and polydactyly occur with increased frequency (*Di Donato et al., 2014*; *Rivière et al., 2012*). Variants in *ACTB* alone are responsible for two other well-defined syndromes. *ACTB*-associated pleiotropic malformation syndrome and *ACTB*-associated syndromic thrombocytopenia can be caused by single-point missense mutations, truncating mutations, frameshift mutations, or deletion (*Latham et al., 2018*; *Cuvertino et al., 2017*). These *ACTB*-associated syndromes differ from BWCFF in that they cause only mild craniofacial malformations and developmental disorders (*Cuvertino et al., 2017*; *Latham et al., 2018*). Selected single-point missense mutations in *ACTG1* lead to a non-syndromic form of hearing loss (*Rendtorff et al., 2006*; *van Wijk et al., 2003*; *Yuan et al., 2016*).

How exactly mutations in the ubiquitously produced actin isoforms of the cytoskeleton lead to various syndromes and mutation-specific symptoms remains unclear. Currently, four main disease mechanisms are under discussion (*Greve et al., 2022*; *Costa et al., 2004*). First, mutations in the actin isoform can lead to functional haploinsufficiency, where the produced molecule is either intrinsically unstable and therefore degraded by the proteasome or unable to associate with other actin molecules and thus unable to partake in the necessary steps involved in the dynamic remodeling of the actin cytoskeleton. This results in a reduced amount of 'active' actin molecules, which cannot fully be compensated for by the remaining allele (*Vang et al., 2005*). Second, the mutant protein is produced in normal quantities and integrated into filaments but interferes with actin filament turnover (*Costa et al., 2004*). Third, the mutation perturbs interactions with ABPs, either by direct or allosteric disturbance of actin–actin or actin–ABP interaction interfaces (*Greve et al., 2022*; *Hundt et al., 2014*; *Bryan et al., 2006*; *Müller et al., 2012*). Fourth, the mutant protein shows an increased propensity for forming toxic rod-like oligomeric structures, which can potentially harm the cell (*Ilkovski et al., 2004*; *Schröder et al., 2004*).

Therefore, one of the significant challenges is elucidating the relationship between the disturbed molecular mechanisms of the cell's actin dynamics and disease patterns observed in patients. In order

to achieve this, it is necessary to resolve the structure–function relationship of the various actin mutations at the molecular level.

Here, we report the biochemical characterization of p.E334Q, a cytoskeletal γ-actin variant, which causes a syndromic form of non-muscle actinopathy. Features associated with the production of p.E334Q include mild facial anomalies and an abnormal gyral pattern of the cerebral cortex (*Di Donato et al., 2016*). At 4 y (age of diagnosis), a patient carrying this mutation showed developmental delay and severe muscle hypotonia (*Di Donato et al., 2016*). Although the patient was initially reported within the *ACTG1*-associated BWCFF cohort, careful reassessment of the phenotype revealed that the clinical phenotype cannot be classified as BWCFF and is therefore assigned as an unclassified NMA.

Structural analysis of the disease-causing mutation shows that residue E334 is located in the hinge region of the actin molecule, which connects the two subdomain pairs of the protein. The region's flexibility and ability to act as a pivot point are essential for the subdomain rearrangement when actin monomers polymerize into filaments (*Figure 1A*). Furthermore, E334 is part of a contact region on the F-actin surface that interacts with the so-called cardiomyopathy loop in the motor domain of cytoskeletal myosin isoforms and contributes to the formation of the actin–cofilin complex (Figure 3).

Here, we present experimental evidence for the impact of changes involving residue E334 with respect to productive interactions with cytoskeletal myosin isoforms, actin polymerization dynamics, and the formation of stable actin–cofilin complexes. We show that mutation E334Q does not adversely affect protein folding. The thermal stability of p.E334Q resembles that of the wild-type protein, both as a monomer and following integration into filaments. The mutant protein efficiently copolymerizes with wildtype (WT) monomers and integrates into linear and branched filament networks. In contrast, the interaction of mutant filaments with non-muscle myosin-2A/C and myosin-5A is strongly impaired, as shown by in vitro motility experiments and solution kinetics. Furthermore, p.E334Q filaments are less susceptible to cofilin-mediated disassembly, while monomers are more susceptible to cofilin-mediated nucleation.

## Results

### Production of cytoskeletal γ-actin and its variant p.E334Q

We produced untagged cytoskeletal γ-actin and its variant p.E334Q in insect cells, free of contaminating endogenous *Spodoptera frugiperda* actin, using a method that was originally designed for the production of toxic actin mutants in *Dictyostelium discoideum* and later adapted for use with the baculovirus/Sf9 insect cell expression system (*Noguchi et al., 2007*; *Lu et al., 2015*). The monomeric actin constructs were initially produced fused to the actin monomer sequestering protein thymosin β4. Purification of the fusion protein by immobilized metal affinity chromatography, followed by chymotrypsin-mediated cleavage of C-terminal linker and tag sequences, results in homogeneous protein without non-native residues and native N-terminal processing, which includes cleavage of the initial methionine and acetylation of the following glutamate. The method worked equally well for both WT and mutant actin. Starting with $2 \times 10^9$ cells, typical yields ranged from 5 to 7 mg of purified protein for cytoskeletal γ-actin and p.E334Q (N = 3 for WT and N = 4 for p.E334Q) (*Figure 1—figure supplement 1*).

### Impact of mutation E334Q on protein folding and thermal stability

Mutation-induced misfolding of actin has been identified as one of the main causes for the development of actinopathies (*Bryan et al., 2006*; *Vang et al., 2005*). Residue E334 is located in the hinge region, formed by residues 137–145 and 333–338 (*Figure 1A*), which is strongly involved in the structural reorganization of the monomer upon incorporation into a growing filament (*Dominguez and Holmes, 2011*). Previous studies have shown that an intact hinge region is crucial for the interactions with the eukaryotic chaperonin CCT/TriC and myosin (*Noguchi et al., 2012*; *McCormack et al., 2001a*; *McCormack et al., 2001b*). We first performed a DNase I inhibition assay to analyze the effect of mutation E334Q on the folding of the protein. Previous studies have shown that the activity of DNase I is only effectively inhibited by correctly folded actin (*Schüler et al., 2000*). We observed a 10% increase in the apparent $IC_{50}$ value of p.E334Q (25.7 ± 0.6 nM) compared to the WT protein (22.7 ± 0.7 nM) (*Figure 1B*). This is in good agreement with circular dichroism (CD) spectroscopy results showing nearly identical secondary structure compositions for WT and mutant proteins

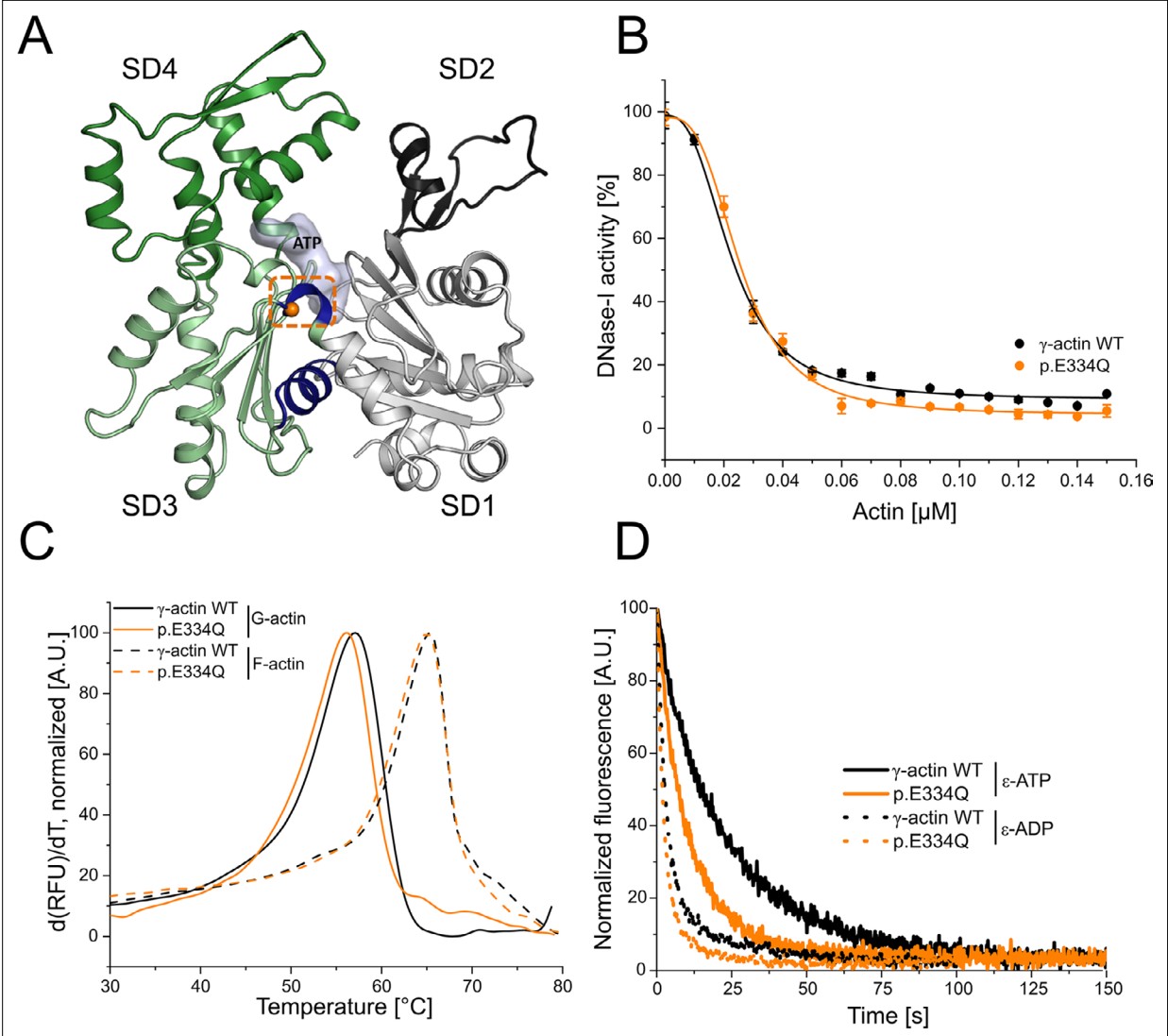

**Figure 1.** Analysis of actin folding, thermal stability, and nucleotide exchange. (**A**) Homology model of human cytoskeletal γ-actin (based on PDB ID: 2BTF). Subdomains (SDs) of actin are colored in gray (SD1), dark gray (SD2), light green (SD3), and green (SD4). The hinge region encompassing residues 137–145 and 333–338 is colored in blue. The mutant residue E334 is shown as an orange sphere. (**B**) Inhibition of DNase I activity by monomeric γ-actin WT and p.E334Q. Data is the mean of three individual experiments ± SD. A Hill equation was fitted to the data, which yields the half-maximal inhibitory concentration (IC$_{50}$, **Table 1**). (**C**) The protein denaturation temperature of monomeric and filamentous p.E334Q and WT actin were determined by differential scanning fluorimetry (DSF). Representative experimental traces are shown. The protein denaturation temperature is derived from the peak of the melting curve (**Table 1**) (p.E334Q [G-actin], N = 3; WT [G-actin], N = 5; p.E334Q [F–actin], N = 5; WT [F–actin], N = 5). (**D**) Nucleotide exchange rates ($k_{-T}$, $k_{-D}$) were determined for monomeric p.E334Q and γ-actin WT using fluorescently labeled ATP (ε-ATP) (p.E334Q, N = 18; WT, N = 21) and ADP (ε-ADP) (p.E334Q, N = 11; WT, N = 11) Representative experimental traces are shown. Rates were determined by fitting a single-exponential function to the data (**Table 1**).

The online version of this article includes the following source data and figure supplement(s) for figure 1:

**Figure supplement 1.** Purification of recombinant WT and p.E334Q γ-actin from *Spodoptera frugiperda* (Sf-9).

**Figure supplement 1—source data 1.** Scan of the SDS gel (left gel) shown in *Figure 1—figure supplement 1*.

**Figure supplement 1—source data 2.** Scan of the SDS gel (right gel) shown in *Figure 1—figure supplement 1*.

**Figure supplement 1—source data 3.** Additional SDS gel of the same protein samples.

**Figure supplement 2.** Circular dichroism (CD) spectra of γ-actin WT and p.E334Q protein.

(*Figure 1—figure supplement 2*). Thermal denaturation temperatures ($T_M$) analyzed by differential scanning fluorimetry (DSF) gave values of 56.77 ± 0.24°C and 56.19 ± 0.30°C for monomeric WT actin and p.E334Q. The values obtained for WT and E334Q filaments were 65.00 ± 0.76°C and 64.83 ± 0.25°C, respectively (*Figure 1C*, *Table 1*). Overall, these results show that the mutant actin is correctly folded and has comparable stability to the WT protein.

As the hinge region is in close proximity to the nucleotide-binding cleft, we examined a potential effect of the mutation on the interaction with nucleotides and the turnover of ATP. Initial tests gave near identical rates of 0.00089 ± 0.00005 s$^{-1}$ and 0.00088 ± 0.00004 s$^{-1}$ for the steady-state turnover of ATP by filaments formed from cytoskeletal γ-actin or the p.E334Q variant (*Table 1*). Next, we determined the nucleotide dissociation rates $k_T$ and $k_D$ by monitoring the decrease in fluorescence amplitude associated with the displacement of ε-ATP or ε-ADP after rapid mixing with a large excess of unlabeled ATP. With p.E334Q-actin, we observed rates of $k_T$ = 0.0972 ± 0.0019 s$^{-1}$ and $k_D$ = 0.3199 ± 0.0027 s$^{-1}$. These rates are about 2-fold and 1.5-fold faster than those observed in experiments performed with WT monomers ($k_T$ = 0.0473 ± 0.0013 s$^{-1}$, $k_D$ = 0.2155 ± 0.0017 s$^{-1}$; *Figure 1D*, *Table 1*). The observed nucleotide dissociation rates for human cytoskeletal γ-actin are in good agreement with the values obtained for a mixture of pyrene-labeled cytoskeletal γ-actin (15%) and β-actin (85%) purified from human platelets in experiments performed at the same ionic strength (*De La Cruz and Sept, 2010*).

## p.E334Q shows normal polymerization behavior and is efficiently integrated into linear and branched actin structures

The regulated assembly and disassembly of filamentous actin drive various cellular processes, including directed cell migration, adhesion, and endocytosis. A slight conformational change of the actin monomer accompanies the integration of a monomeric actin molecule into a growing filament. A small change in the hinge region leading to a 12° to 13° rotation of the inner subdomain pair relative to the outer subdomain pair results in a flatter actin molecule within the filament compared to the free actin monomer (*Dominguez and Holmes, 2011*; *Oda et al., 2009*). Mutations in the hinge region can thus potentially affect filament assembly by impairing the required structural reorganization of the monomer as it attaches to the growing filament end. Total internal reflection fluorescence (TIRF) microscopy-based tracking of the polymerization of monomeric Atto 655-labeled WT or mutant actin in vitro revealed a small but significant reduction in the elongation rate from 12.6 ± 0.3 nm s$^{-1}$ for WT to 11.1 ± 0.3 nm s$^{-1}$ for mutant filaments (*Figure 2A and B*). To detect any differences in nucleation efficiency, we determined the time-dependent increase in the number of filaments (*Figure 2C*). Aside from short-term fluctuations, the linear increase in the number of filaments recorded for WT (0.039 ± 0.002 s$^{-1}$) and p.E334Q (0.041 ± 0.004 s$^{-1}$) was similar over the full 1000s duration of the experiment. Overall, p.E334Q shows only a mild polymerization defect.

Since actin polymerization in vivo is tightly regulated in a spatio-temporal manner by different ABPs, we analyzed the interaction of p.E334Q with one of the key regulators of actin polymerization, the Arp2/3 complex (*Goley and Welch, 2006*). The Arp2/3 complex promotes the nucleation of new actin filaments from the sides of existing filaments and caps their pointed ends. Thereby, it promotes the formation of branched actin networks and contributes to essential processes that depend on the dynamic remodeling of the actin cytoskeleton, such as cell migration, endocytosis, and organelle positioning (*Nicholson-Dykstra et al., 2005*; *Mullins et al., 1998*; *Blanchoin et al., 2000*; *Loisel et al., 1999*). The ability of p.E334Q to form branched actin networks was probed in TIRF microscopy-based experiments in the presence of increasing concentrations of VCA-activated human Arp2/3 complex (*Figure 2D*). At the lowest Arp2/3 concentration used (0.01 nM), we observed the appearance of the first branches after approximately 250 s. In the presence of 0.1 nM and 0.2 nM of the VCA-activated Arp2/3 complex, mutant and WT actin continued forming branched networks with comparable branch formation and branch elongation rates. Similar results were obtained with WT actin, mutant actin, or with 1:1 mixtures of WT and mutant actin, when we analyzed the effect of adding increasing concentrations of VCA-activated human Arp2/3 in pyrene actin-based bulk polymerization assays (*Figure 2E and F*). The results are summarized in *Table 1*.

**Table 1.** Comparison of biochemical parameters, velocities, rate, and equilibrium constants measured in experiments with human cytoskeletal $\gamma$-actin and variant p.E334Q.

| Parameter | $\gamma$-actin | p.E334Q | 1:1 mixture |
|---|---|---|---|
| IC$_{50}$ (nM) (DNase inhibition) | 22.7 ± 0.7 | 25.7 ± 0.6 (**\*1.1-fold ↑**) | – |
| $T_M$ (°C) (Mg$^{2+}$-ATP-G-actin) | 56.77 ± 0.24 | 56.19 ± 0.30 (**ns**) | – |
| $T_M$ (°C) (Mg$^{2+}$-ADP-F-actin) | 65.00 ± 0.76 | 64.83 ± 0.25 (**ns**) | – |
| $k_T$ (s$^{-1}$) (Mg$^{2+}$-ε-ATP) | 0.0473 ± 0.0013 | 0.0972 ± 0.0019 (**\*\*\*\* 2.1-fold ↑**) | – |
| $k_D$ (s$^{-1}$) (Mg$^{2+}$-ε-ADP) | 0.2155 ± 0.0017 | 0.3199 ± 0.0027 (**\*\*\*\* 1.5-fold ↑**) | – |
| F-actin ATPase (s$^{-1}$) | 0.00089 ± 0.00005 | 0.00088 ± 0.00004 (**ns**) | – |
| $k_{obs}$ (s$^{-1}$) bulk polymerization rate (Δpyr-fluorescence) | 0.0140 ± 0.0004 | 0.0129 ± 0.0008 (**\*\*\*1.1-fold ↓**) | 0.0134 ± 0.0006 (**\*1.04-fold ↓**) |
| $k_{obs}$ (s$^{-1}$) + 10 nM Arp2/3 bulk polymerization rate (Δpyr-fluorescence) | 0.6441 ± 0.0163 | 0.5458 ± 0.02256 (**\*\*1.2-fold ↓**) | 0.6162 ± 0.0776 (**ns**) |
| $r_{obs}$ (nm s$^{-1}$) rate of filament elongation (TIRF-M) | 12.6 ± 0.3 | 11.1 ± 0.3 (**\*\*\*\* 1.1-fold ↓**) | – |
| $k_{obs}$ (s$^{-1}$) rate of filament disassembly (Δpyr-fluorescence) | 0.0022 ± 0.0002 | 0.0029 ± 0.0003 (**\*\*\*\* 1.3-fold ↑**) | 0.0023 ± 0.0002 (**ns**) |
| $k_c$ (nM$^{-1}$ s$^{-1}$) + cofilin rate of filament disassembly (Δpyr-fluorescence) | $3.42 \times 10^{-4}$ ± $0.22 \times 10^{-4}$ | $0.81 \times 10^{-4}$ ± $0.08 \times 10^{-4}$ (**\*\*\*\* 4.2-fold ↓**) | $1.54 \times 10^{-4}$ ± $0.11 \times 10^{-4}$ (**\*\*\* 2.2-fold ↓**) |
| v (nm s$^{-1}$) sliding velocity on NM2A-HMM | 195.3 ± 5.0 | 39.1 ± 3.2 (**\*\*\*\* 5.0-fold ↓**) | 131.2 ± 10.0 (**\*\*\* 1.5-fold ↓**) |
| v (nm s$^{-1}$) sliding velocity on Myo5A-HMM at optimal myosin density | 315.1 ± 28.9 | 190.4 ± 46.9 (**\*\*\*\* 1.7-fold ↓**) | – |
| $K_{app}$ (µM) (NM2A-HMM) | 3.20 ± 0.74 | 2.89 ± 0.49 (**ns**) | – |
| $k_{cat}$ (s$^{-1}$) (NM2A-HMM) | 0.097 ± 0.002 | 0.076 ± 0.005 ( **\*1.3-fold ↓**) | – |
| $k_A$ (s$^{-1}$) (NM2A-2R) | $8.0 \times 10^{-5}$ ± $3.3 \times 10^{-6}$ | $4.5 \times 10^{-4}$ ± $6.7 \times 10^{-6}$ (**\*\*\*\* 5.6-fold ↑**) | $2.3 \times 10^{-4}$ ± $1.3 \times 10^{-5}$ (**\*\*\*2.9-fold ↑**) |
| $k_A$ (s$^{-1}$) (NM2C-2R) | $3.1 \times 10^{-4}$ ± $5.0 \times 10^{-6}$ | $1.7 \times 10^{-3}$ ± $5.8 \times 10^{-5}$ (**\*\*\*5.6-fold ↑**) | $6.7 \times 10^{-4}$ ± $1.5 \times 10^{-5}$ (**\*\*\*2.2-fold ↑**) |
| $K_A$ (nM) (NM2A-2R) | 1.7 ± 0.3 | 16.9 ± 0.3 (**\*\*9.9-fold ↑**) | – |
| $K_{DA}$ (nM) (NM2A-2R) | 122.6 ± 11.4 | 353.0 ± 20.6 (**\*\*2.9-fold ↑**) | – |
| Coupling ($K_{DA}/K_A$) (NM2A-2R) | 72.1 | 20.9 (**\*\*\*\* 3.5-fold ↓**) | – |
| $1/K_1$ (µM) (NM2A-2R) | 812.1 ± 82.7 | 863.8 ± 91.6 (**ns**) | – |
| $k_{+2}$ (s$^{-1}$) (NM2A-2R) | 461.7 ± 18.6 | 435.6 ± 16.5 (**ns**) | – |
| $K_1k_{+2}$ (µM$^{-1}$ s$^{-1}$) (NM2A-2R) | 0.276 ± 0.002 | 0.272 ± 0.005 (**ns**) | – |

Bold entries indicate the observed difference between wt and mutant values. ns = p>0.05, * p≤0.05 , ** p≤0.01, *** p≤0.001, **** p≤0.0001

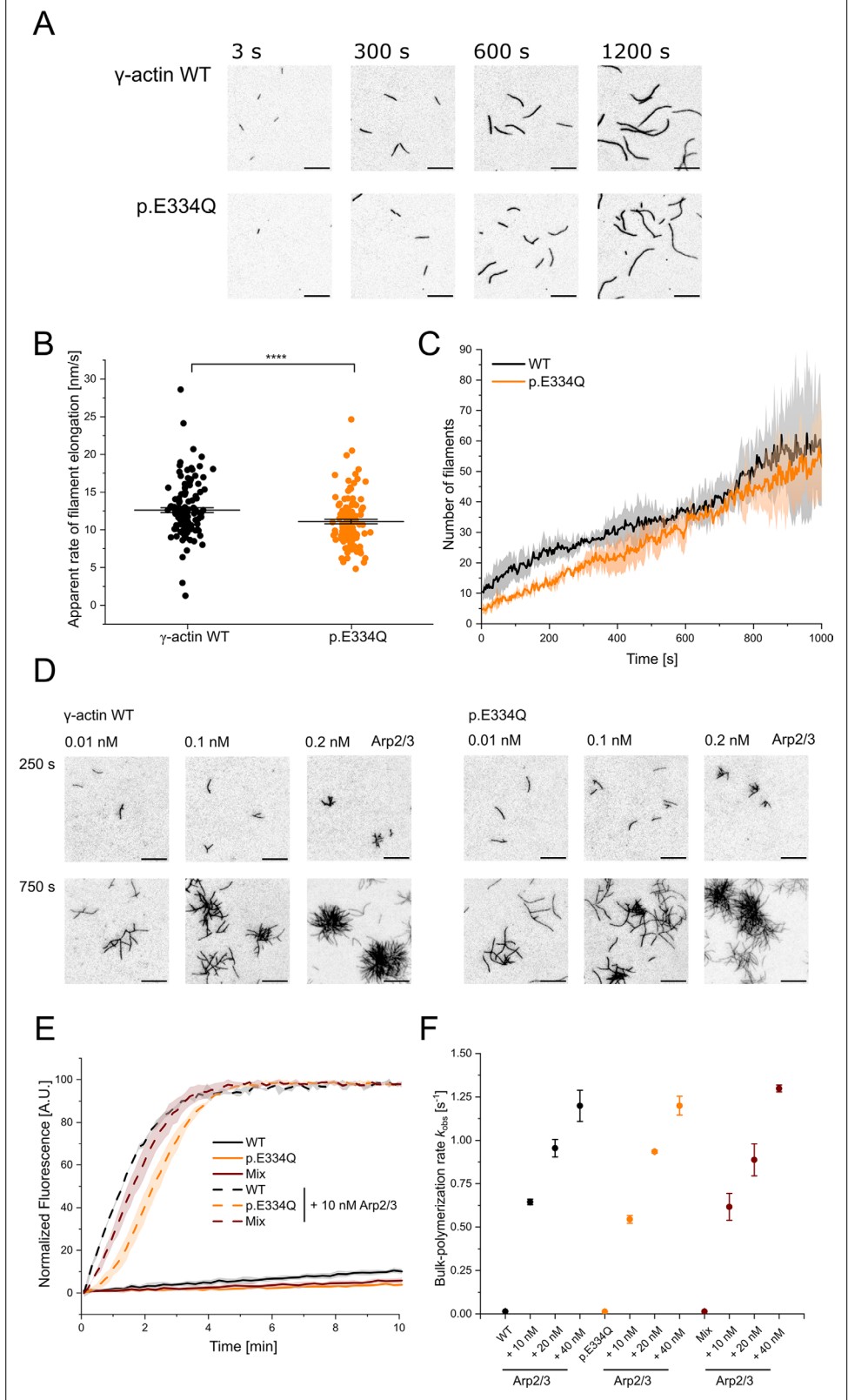

**Figure 2.** Analysis of the polymerization capacity of WT and p.E334Q γ-actin in the absence and presence of human Arp2/3 complex. (**A**) Polymerization of p.E334Q and WT γ-actin (1 μM, 10% Atto 655 labeled) was induced by salt shift and the progression of the reaction was tracked by total internal reflection fluorescence (TIRF) microscopy. Shown are representative micrographs at the indicated points in time. Scale bar corresponds

*Figure 2 continued on next page*

*Figure 2 continued*

to 10 μm. (**B**) The elongation rates of individual filaments were determined by manual tracking of the elongating barbed ends of the filaments. Every data point represents an individual filament (WT: n = 128 filaments; p.E334Q: n = 114 filaments). Data is shown as the mean ± SEM. (**C**) Nucleation efficiencies were determined by monitoring changes in the number of filaments over the time of the polymerization experiments. The solid lines and shades represent the mean ± SD of four individual experiments. (**D**) TIRF microscopy-based observation of human Arp2/3 complex-mediated differences in the salt-induced polymerization of p.E334Q and WT γ-actin (1 μM, 10% Atto 655 labeled). Shown are representative micrographs at the indicated points in time. For both WT and p.E334Q γ-actin, the formation of the first branch points was observed after 250 s in the presence of 0.01 nM Arp2/3. Scale bar corresponds to 10 μm. N = 3 for each Arp2/3 concentration .(**E**) Pyrene-based polymerization experiment of 2 μM γ-actin WT, p.E334Q actin, or a 1:1 mixture of both (5% pyrene labeled) in the absence and presence of Arp2/3. Shown are representative traces. The solid lines/shades represent the mean ± SD of at least three individual experiments. (**F**) Bulk polymerization rates determined from the experiments shown in (**E**). Data is shown as the mean ± SD of all performed experiments. N = 3 for each Arp2/3 concentration. N = 12 for all experiments performed in the absence of Arp2/3. Individual $k_{obs}$ values and differences with statistical significance are summarized in *Table 1*.

## Impact of E334Q mutation on interactions with ADF/cofilin family proteins

Homology models of the complexes formed by human cofilin-1 with filamentous cytoskeletal γ-actin and p.E334Q were generated to evaluate the functional impact of mutation E334Q. Cofilin-1 binds to actin filaments by making simultaneous contact with two adjacent monomers. Actin residue E334 is located halfway between two adjacent cofilin molecules, but only makes direct contact with residue L111 of the cofilin-1 molecule on the pointed end side (*Figure 3A*). Cofilin-1 residue L111 is located in an α-helix (L111-L128) that is essential for actin binding (*Pope et al., 2000*). Mutation E334Q maintains the van der Waals interaction with L111, but reduces the electrostatic attraction between the interacting surfaces, with likely effects on the stability and allosteric coupling of the complex (*Figure 3B and C*).

To evaluate the impact of mutation E334Q on the elastic properties of F-actin in the presence and absence of cofilin-1 binding, we performed normal mode analysis using bare, partially decorated, and fully decorated filaments (*Figure 3*, *Figure 3—figure supplement 1A*). Since residue E334 is located on the outer surface of the actin filament, the mutation produces no significant change in the normal mode frequencies of bare filaments. Similarly, fully decorated filaments show only minor differences. Substantial changes in the low-frequency modes associated with global conformational changes were observed when comparing the partially cofilin-decorated filaments of WT and mutant filaments, where the bare and cofilin-decorated regions move as rigid bodies and the interface between the two regions acts as a hinge (modes 7, 8) or fulcrum (mode 9) (*Figure 3—figure supplement 1B*).

To test the validity of our modeling results, we performed biochemical and functional assays probing the interactions of human cofilin-1 with monomeric and filamentous cytoskeletal γ-actin and p.E334Q. The effect of cofilin-1 on filament assembly was assessed by TIRF microscopy experiments. The polymerization of 1 μM actin was induced in the presence of 0, 50, or 100 nM cofilin-1. Following an initial lag phase of 200–300 s, the filament formation rate increased with the cofilin-1 concentration (*Figure 4*). In experiments with p.E334Q-actin, de novo filament nucleation proceeded faster (*Figure 4A and B*). Movies showing filament formation by de novo nucleation in the presence of cofilin-1 are available in *Figure 4—videos 1–6*.

Next, we investigated the effect of cofilin-1 on the disassembly of preformed cytoskeletal γ-actin and p.E334Q filaments. The effect of cofilin-1 on existing WT and mutant filaments was initially tested by performing pyrene-based dilution-induced disassembly experiments with actin filaments polymerized for 16 hr and containing predominately ADP actin. Cofilin-1 has previously been shown to increase the bulk disassembly rate of actin in this experimental setup (*Breitsprecher et al., 2011*; *Greve et al., 2022*). In the absence of cofilin, the changes in pyrene fluorescence are best described by single exponential functions with $k_{obs}$ values of 0.0022 ± 0.0002 s$^{-1}$ and 0.0029 ± 0.0003 s$^{-1}$ for WT and p.E334Q filaments, respectively (*Figure 5A*, *Figure 5—figure supplement 1*). The addition of cofilin-1 increased the observed rate of the dilution-induced disassembly in a concentration-dependent manner for both mutant and WT filaments. At the highest cofilin-1 concentration used (40 nM), the change in pyrene fluorescence with p.E334Q filaments is still best described by a single

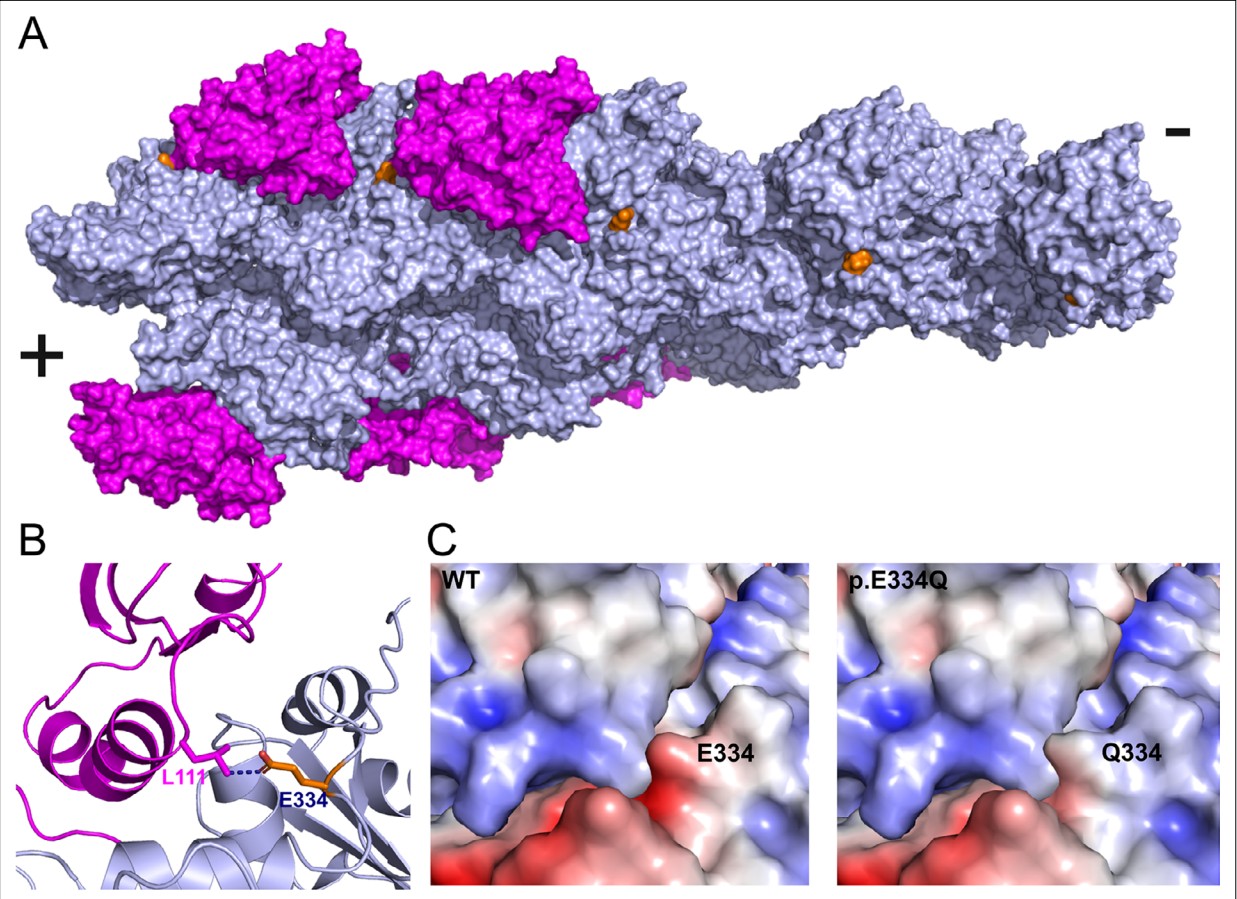

**Figure 3.** In silico analysis of the effects of the E334Q mutation on the actin–cofilin interaction. (**A**) Human homology model of a cytoskeletal γ-actin filament partially decorated with cofilin-1 (model based on PDB ID: 6UC4). Cofilin-1 is colored magenta, actin monomers are depicted in light blue, and the actin residue E334 is colored orange. (**B**) Closeup showing the details of the van der Waals interaction between E334 and cofilin residue L111 on the pointed end side. The length of the dotted line between the side chains of actin-E334 and cofilin-L111 corresponds to 3.1 Å in our model. (**C**) The change in surface charge at the actin–cofilin interaction interface introduced by mutation E334Q is visualized by coulombic surface coloring.

The online version of this article includes the following figure supplement(s) for figure 3:

**Figure supplement 1.** Normal mode analysis of changes induced by mutation E334Q in the global mobility of bare, partially, and a fully cofilin-1 decorated γ-actin mini-filaments.

exponential function with a $k_{obs}$ value of 0.0059 ± 0.0008 s⁻¹. In contrast, the change for WT filaments is best fitted to a bi-exponential function. Here, the $k_{obs}$ values for the fast and slow phases are 0.0163 ± 0.0028 s⁻¹ and 0.0002 ± 0.00003 s⁻¹, respectively (*Figure 5A*, *Figure 5—figure supplement 1*). Additional experiments with heterofilaments showed milder but significant differences (*Table 1*). The observed rate constant values were linearly dependent on the concentration of cofilin-1 in the range 0–40 nM, with the slopes corresponding to the apparent second-order rate constant ($k_C$) for the cofilin-1-induced depolymerization of F-actin. In experiments performed with p.E334Q filaments, the value obtained for $k_C$ was 4.2-fold lower (0.81 × 10⁻⁴ ± 0.08 × 10⁻⁴ nM⁻¹ s⁻¹) compared to experiments with WT filaments (3.42 × 10⁻⁴ ± 0.22 × 10⁻⁴ nM⁻¹ s⁻¹). When heterofilaments were used, the effect of the mutation was reduced to a 2.2-fold difference compared to WT filaments (1.54 × 10⁻⁴ ± 0.11 × 10⁻⁴ nM⁻¹ s⁻¹).

The pyrene-based bulk experiments show an evident change in the p.E334Q–cofilin interaction. However, they cannot distinguish between cofilin-1 binding defects and a disassembly efficiency change. To analyze the perturbed interaction of p.E334Q with cofilin-1 in more detail, we first performed high-speed co-sedimentation assays with constant concentrations of F-actin (5 µM) and increasing concentrations of cofilin-1 (0.0625–20 µM) at pH 6.5 and 7.8. Cofilin-1 binds faster but severs filaments less efficiently under slightly acidic pH conditions, whereas a weakly basic environment

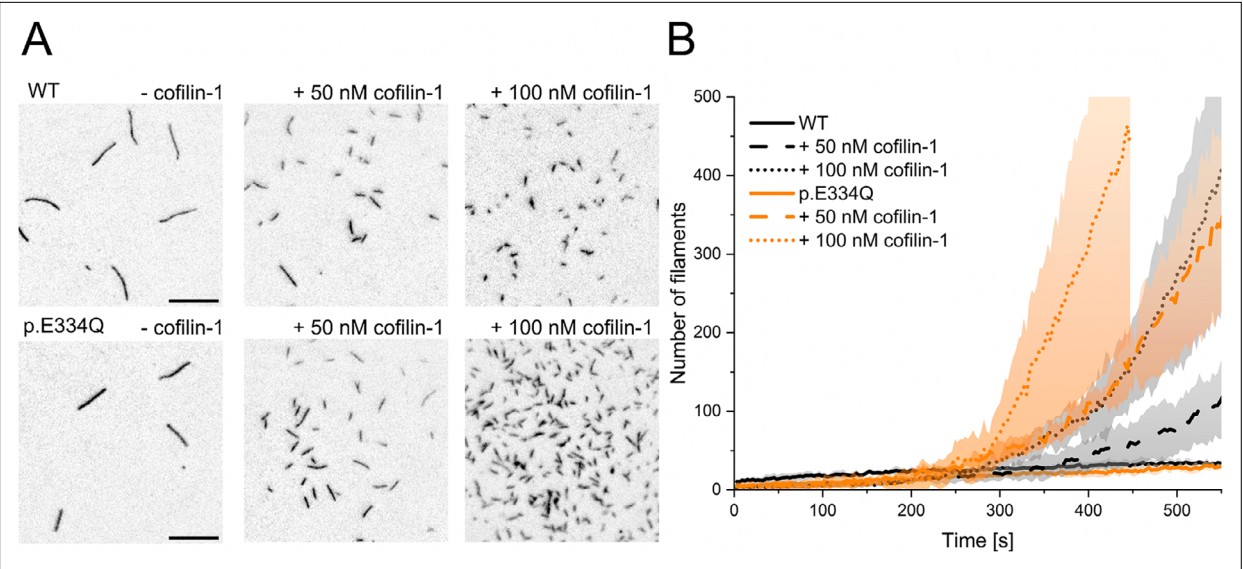

**Figure 4.** Analysis of the effect of cofilin-1 on the spontaneous polymerization of WT and p.E334Q γ-actin. (**A**) Total internal reflection fluorescence (TIRF) microscopy-based observation of cofilin-1-mediated differences in the salt-induced polymerization of p.E334Q and WT γ-actin (1 μM, 10% Atto 655 labeled). Shown are representative micrographs 450 s after induction of polymerization. Scale bar corresponds to 10 μm. (**B**) Time dependence of the increase in filaments observed by TIRF microscopy. The solid and dotted lines and shades represent the mean ± SD of three individual experiments.

The online version of this article includes the following video(s) for figure 4:

**Figure 4—video 1.** Polymerization of 1 μM WT γ-actin (10% Atto 655 labeled) visualized by total internal reflection fluorescence (TIRF) microscopy.
https://elifesciences.org/articles/93013/figures#fig4video1

**Figure 4—video 2.** Polymerization of 1 μM WT γ-actin (10% Atto 655 labeled) in the presence of 50 nM cofilin-1 visualized by total internal reflection fluorescence (TIRF) microscopy.
https://elifesciences.org/articles/93013/figures#fig4video2

**Figure 4—video 3.** Polymerization of 1 μM WT γ-actin (10% Atto 655 labeled) in the presence of 100 nM cofilin-1 visualized by total internal reflection fluorescence (TIRF) microscopy.
https://elifesciences.org/articles/93013/figures#fig4video3

**Figure 4—video 4.** Polymerization of 1 μM p.E334Q γ-actin (10% Atto 655 labeled) visualized by total internal reflection fluorescence (TIRF) microscopy.
https://elifesciences.org/articles/93013/figures#fig4video4

**Figure 4—video 5.** Polymerization of 1 μM p.E334Q γ-actin (10% Atto 655 labeled) in the presence of 50 nM cofilin-1 visualized by total internal reflection fluorescence (TIRF) microscopy.
https://elifesciences.org/articles/93013/figures#fig4video5

**Figure 4—video 6.** Polymerization of 1 μM p.E334Q γ-actin (10% Atto 655 labeled) in the presence of 100 nM cofilin-1 visualized by total internal reflection fluorescence (TIRF) microscopy.
https://elifesciences.org/articles/93013/figures#fig4video6

favors cofilin-mediated severing (*Wioland et al., 2019*). We first performed experiments at pH 6.5. This endpoint experiment showed that cofilin-1 binds to and disassembles p.E334Q and WT filaments (*Figure 5B and C*). Most pellet fractions from the experiments conducted with WT filaments were found to contain less cofilin-1 as higher disassembly efficiency results in more cofilin-1 being bound to actin monomers and retained in the supernatant fraction. We repeated the experiment at pH 7.8 with an equimolar and a twofold excess of cofilin (*Figure 5B and C*). In experiments with equimolar cofilin-1, we found significantly less WT actin in the pellet fraction than at pH 6.5, consistent with a higher disassembly efficiency at pH 7.8. The shift to pH 7.8 did not have the same enhancing effect on the disassembly of p.E334Q filaments, as indicated by an increase in p.E334Q actin in the pellet fraction. The endpoint measurements show that cofilin-1 binds to p.E334Q filaments but severs them less efficiently. Next, we performed fluorescence quench experiments with pyrene-labeled WT and p.E334Q filaments as it was previously shown that cofilin-1 efficiently quenches the fluorescence of pyrene-labeled actin filaments (*Blanchoin and Pollard, 1999*; *De La Cruz, 2005*). We determined the change in fluorescence amplitude when 2 μM pyrene-labeled p.E334Q or WT filaments were mixed

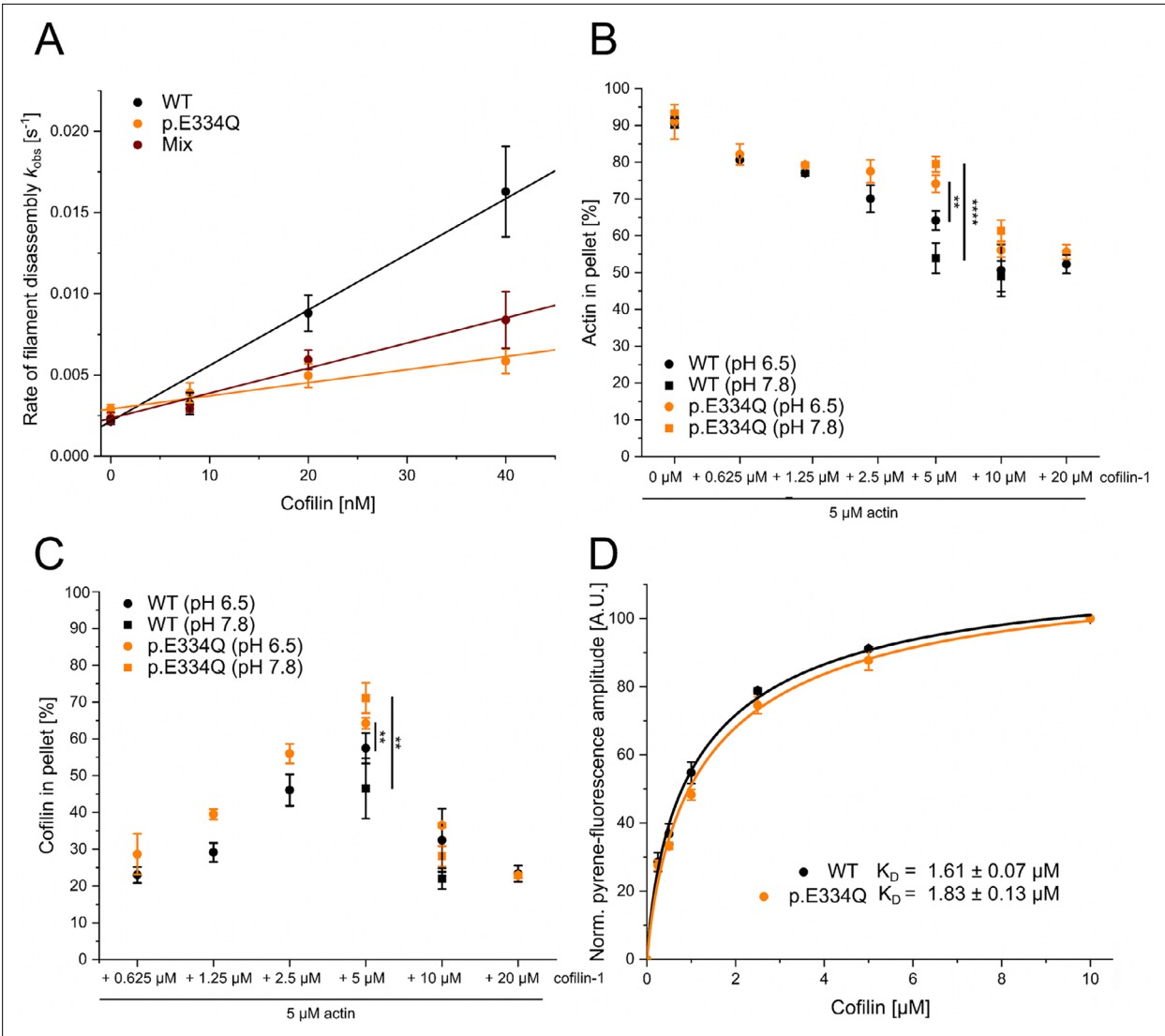

**Figure 5.** Analysis of the interaction of filamentous WT and p.E334Q γ-actin with cofilin-1. (**A**) Pyrene-based dilution-induced depolymerization experiments (5% pyrene-labeled actin) were used to quantify the impact of cofilin-1 on the disassembly of filaments formed by p.E334Q, WT, and a 1:1 mixture of p.E334Q and WT γ-actin. Depolymerization was induced by adding G-buffer (pH 7.8). Representative experimental traces showing the decrease in pyrenyl fluorescence over time are given in *Figure 5—figure supplement 1*. The observed values of the rate constants show a linear dependence on the cofilin-1 concentration in the range of 0–40 nM, with the slopes corresponding to the apparent second-order rate constant ($k_C$) for cofilin-1-induced depolymerization. Values for $k_C$ are summarized in *Table 1*. Data is shown as the mean ± SD of 8–14 individual experiments. (**B, C**) In addition, cofilin-1 binding to actin filaments and cofilin-1-mediated filament disassembly at different pH values (pH 6.5, pH 7.8) was investigated in co–sedimentation experiments. Data is shown as mean ± SD. N = 6 for experiments performed at pH 7.8. N = 7 for experiments performed at pH 6.5 in the absence of cofilin. N = 3 for experiments performed at pH 6.5 in the presence of cofilin. (**D**) The dissociation equilibrium constant ($K_D$) for binding of cofilin-1 to pyrene-labeled WT or p.E334Q filaments was determined by monitoring the change in fluorescence amplitude upon cofilin-1 binding. Data is shown as the mean ± SD. N = 8 for WT, N = 9 for p.E334Q.

The online version of this article includes the following source data and figure supplement(s) for figure 5:

**Figure supplement 1.** Exemplary traces of dilution-induced depolymerization experiments in the absence and presence of increasing concentrations of human cofilin-1.

**Figure supplement 2.** Exemplary polyacrylamide gel showing a co-sedimentation experiment performed with 5 µM WT and p.E334Q γ-actin in the absence and presence of 5 µM or 10 µM cofilin-1 at pH 7.8 (see also: *Figure 5—figure supplement 2—source data 1*).

**Figure supplement 2—source data 1.** Scan of the SDS gel shown in *Figure 5—figure supplement 2*.

with increasing concentrations of cofilin-1. Fitting a quadratic equation to the obtained experimental values gave only slightly different dissociation equilibrium constants of $K_{D,\ p.E334Q}$ = 1.83 ± 0.13 µM and $K_{D,\ WT}$ = 1.61 ± 0.07 µM (*Figure 5D*). These $K_D$ values are in the range of values previously determined for muscle α-actin filaments (*Blanchoin and Pollard, 1999*; *De La Cruz, 2005*), actin filaments from *Acanthamoeba castellanii* (*Blanchoin and Pollard, 1999*), and filaments containing a mixture of human γ- and β-actin (*De La Cruz, 2005*) using the same method.

To better understand the underlying changes leading to the differences in cofilin-mediated filament disassembly efficiency, we performed two-color TIRF microscopy experiments using Atto 655-labeled WT or mutant filaments and eGFP-cofilin-1. This setup allows us to observe the binding of eGFP-cofilin-1 to individual filaments and the resulting disassembly of the filaments in real time. eGFP–cofilin-1 was shown to bind to actin filaments and disassemble actin as efficiently as WT cofilin-1 (*Wioland et al., 2019*; *Wioland et al., 2017*; *Breitsprecher et al., 2011*). Immediately after mixing 300 nM 'aged' actin filaments, capped at their barbed end by human capping protein (CP), with 100 nM eGFP-cofilin-1, the reaction mixture was flushed into a flow chamber and imaging was started. The delay between mixing and the start of imaging was approximately 15 s for each sample. At the end of the experimental dead time, almost twice as many cofilin clusters had formed in the field of view on WT filaments (151 ± 29) as on p.E334Q filaments (82 ± 6) (*Figure 6A*), the number of both WT and mutant actin filaments had doubled from initially about 95 to about 200 filaments (*Figure 6B*), and the Atto 655 fluorescence of the WT filaments had decreased to 30% of the starting value, whereas the Atto 655 fluorescence of p.E334Q filaments remained unchanged (*Figure 6C*). During the following 500 s, total eGFP-cofilin-1 fluorescence increased linearly, with the rate of increase being approximately twice as fast for WT as for p.E334Q filaments (*Figure 6D*). While the total eGFP-cofilin-1 fluorescence in the presence of p.E334Q filaments continued to increase linearly over the next 1000s, the signal for WT filaments began to saturate and reached a plateau value (*Figure 6D*). The number of cofilin clusters on WT filaments decreased steadily during the observation period of 1500s. In contrast, their number increased in the presence of p.E334Q filaments, reaching a maximum after 500 s, followed by a plateau of 400 s and a subsequent slow decrease (*Figure 6A*). The changes in the number of actin filaments showed similar time dependencies, a steady decrease for WT filaments, and an initial increase for p.E334Q filaments, followed immediately by a linear decrease without an intervening plateau phase (*Figure 6B*). For WT and p.E334Q filaments, the size of eGFP-cofilin-1 clusters on the steadily shrinking filaments expanded continuously until either severing occurred or the actin filament was fully decorated. No severing events were observed with fully decorated filaments.

## Impact of E334Q mutation on interactions with myosin family proteins

The precise nature of the six actin-binding elements in the myosin motor domain varies considerably among myosin isoforms (*Sellers, 2000*; *Cope et al., 1996*). The associated sequences have conserved hydrophobic and electrostatic properties, variations of which contribute to the tuning of kinetic and functional properties. Actin residue E334 interacts with a positively charged patch in the cardiomyopathy loop (CM-loop) of the myosin upper 50 kDa domain of NM2C and Myo5A (*Pospich et al., 2021*; *von der Ecken et al., 2016*). In the case of NM2C, the electrostatic interaction of E334 with K429 in the CM loop is further stabilized by van der Waals interactions with I420 and V427 (*Figure 7*). A lysine residue corresponding to K429 in NM2C is present in all human myosin-2 and myosin-5 isoforms. During force-generating interactions with mutant actin filaments, the weakening of complementary electrostatic interactions in this region of the actomyosin interface is likely to attenuate the binding affinity of class 2 and 5 myosins as well as the coupling between their actin and nucleotide-binding sites.

Cytoskeletal γ-actin filaments move with an average sliding velocity of 195.3 ± 5.0 nm s⁻¹ on lawns of surface-immobilized NM2A-HMM molecules (*Figure 8A and B*). For NM2A-HMM densities below about 10,000 molecules per µm², the average sliding speed for cytoskeletal actin filaments drops steeply (*Hundt et al., 2016*). Filaments formed by p.E334Q actin move fivefold slower, resulting in an observed average sliding velocity of 39.1 ± 3.2 nm s⁻¹. Filaments copolymerized from a 1:1 mixture of WT and p.E334Q actin move with an average sliding velocity of 131.2 ± 10 nm s⁻¹ (*Figure 8A and B*). When equal densities of surface-attached WT and mutant filaments were used, we observed that the number of rapid dissociation and association events increased markedly for p.E334Q filaments (*Figure 8—videos 1–3*).

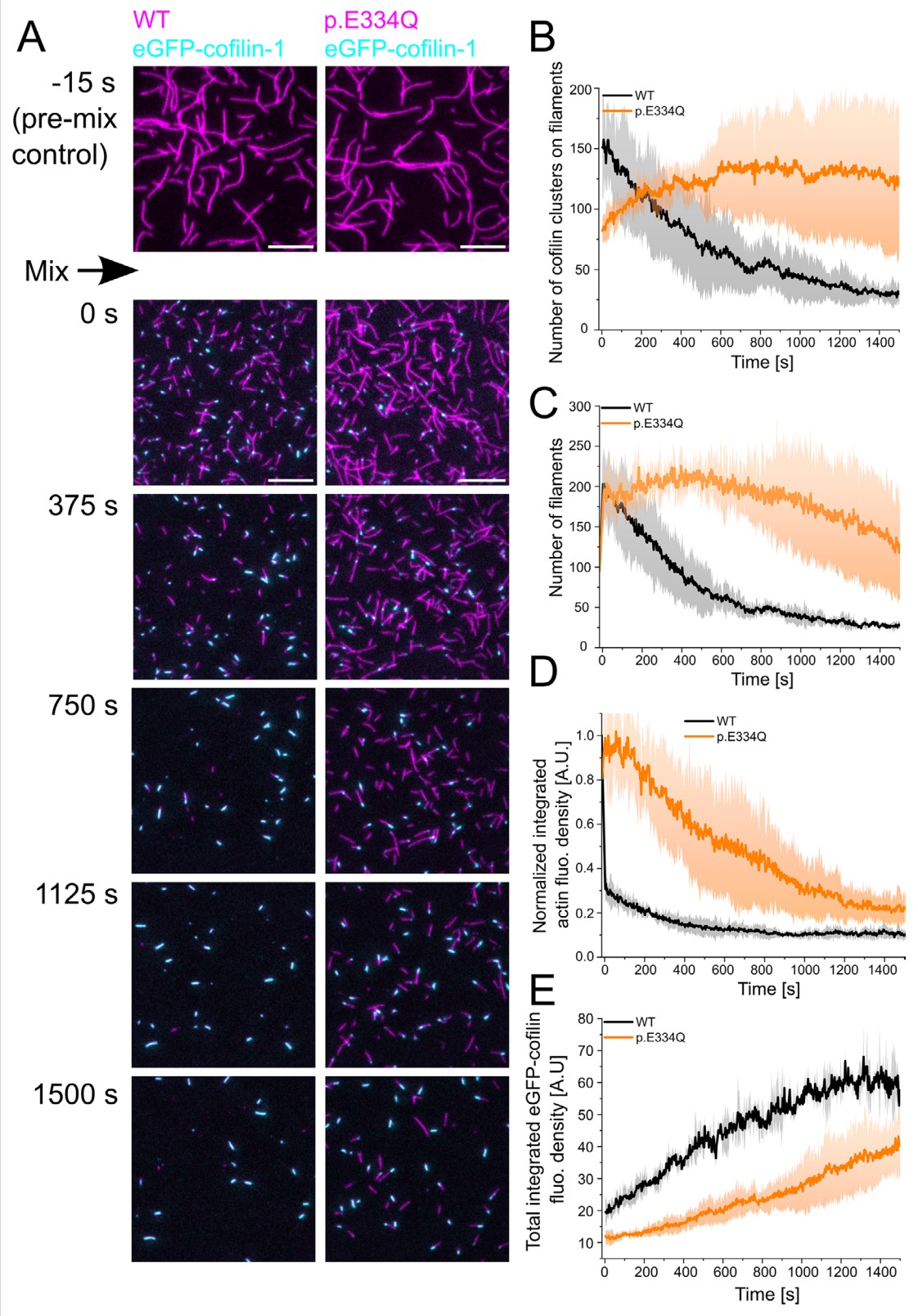

**Figure 6.** Analysis of the interaction of WT and p.E334Q γ-actin with eGFP-cofilin-1. (**A**) The disassembly of 300 nM 'aged' WT and p.E334Q F-actin (10% Atto 655 labeled, capped by CP) in the presence of 100 nM eGFP-cofilin-1 was visualized using two-color total internal reflection fluorescence (TIRF) microscopy. F-actin is shown in magenta, eGFP-cofilin in cyan. Shown are representative micrographs at the indicated time points. Scale bar corresponds to 10 μm. (**B**) Time dependence of the change in the number of eGFP-cofilin-1 clusters on actin filaments. The solid lines and shades

*Figure 6 continued on next page*

*Figure 6 continued*

represent the mean ± SD of three individual experiments. (**C**) Time dependence of the decrease in filament count. The solid lines and shades represent the mean ± SD of three individual experiments. (**D**) Temporal tracking of actin filament disassembly by monitoring changes in total actin fluorescence per image frame. The solid lines and shades represent the mean ± SD of three individual experiments. (**E**) Visualization of temporal changes in the abundance of eGFP-cofilin-1 clusters based on monitoring total eGFP-cofilin-1 fluorescence per image frame. The solid lines and shades represent the mean ± SD of three individual experiments.

The online version of this article includes the following figure supplement(s) for figure 6:

**Figure supplement 1.** Control experiments showing that E334Q and WT γ-actin filaments are stable over a period of 1000s.

Using a NADH-coupled enzymatic assay, we determined the ability of p.E334Q and WT filaments to activate the ATPase of NM2A-HMM over the range of 0–25 µM F-actin (*Figure 8C*). While we observed no significant difference in $K_{app}$, indicated by the actin concentration at half-maximal activation, in experiments with p.E334Q filaments (2.89 ± 0.49 µM) and WT filaments (3.20 ± 0.74 µM), we observed a 28% slower maximal ATP turnover at saturating actin concentration ($k_{cat}$) with p.E334Q filaments (0.076 ± 0.005 s⁻¹ vs 0.097 ± 0.002 s⁻¹). To investigate the impact of the mutation on actomyosin affinity using transient kinetic approaches, we determined the dissociation rate constants using a single-headed NM2A-2R construct (*Figure 8D*). NM2A–2 R is a constitutively active motor–domain construct generated by fusing the motor–domain of NM2A to two spectrin–like repeats serving as a rigid lever arm extension (*Anson et al., 1996*). The construct is ideally suited for the kinetic characterization of the NM2–isoforms (*Heissler and Manstein, 2011*). The dissociation rate constant $k_{-A}$ is 5.6–fold increased from 0.08×10⁻³ s⁻¹ to 0.45×10⁻³ s⁻¹ for NM2A–2 R when p.E334Q filaments are

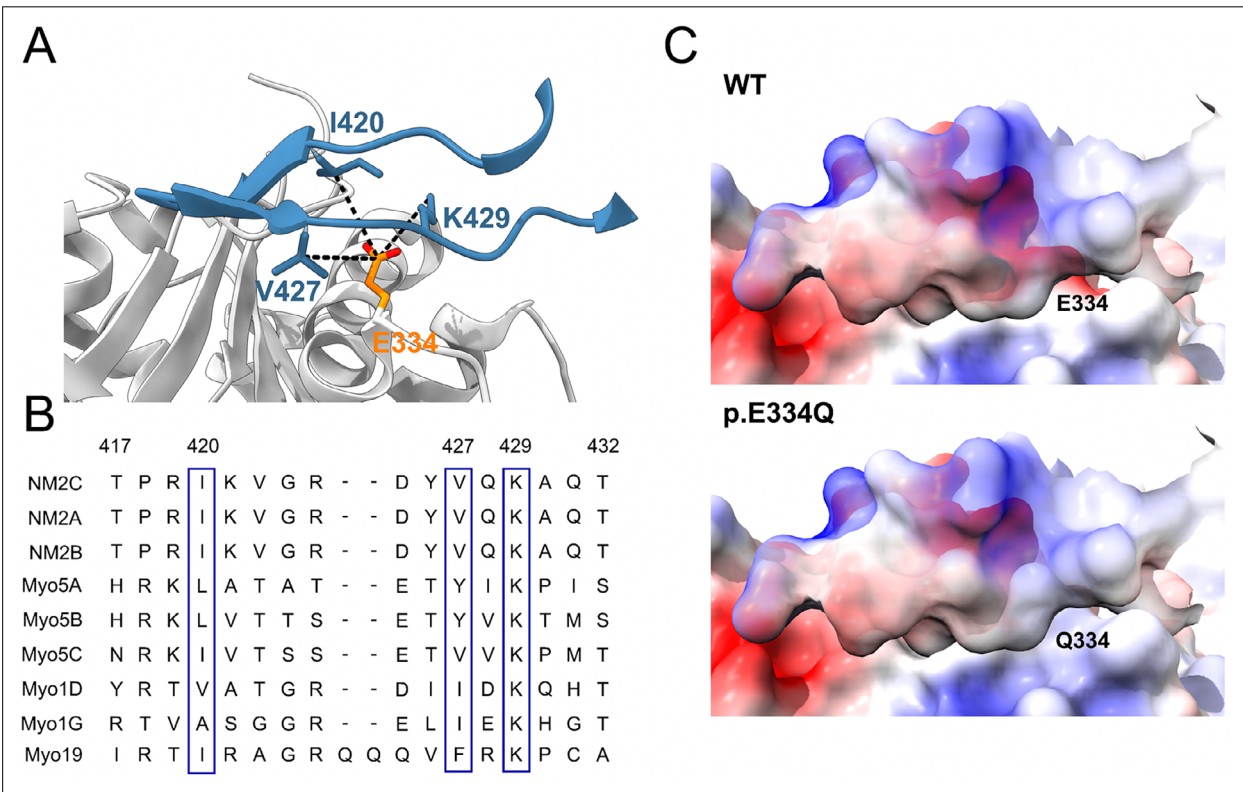

**Figure 7.** In silico analysis of the effects of the E334Q mutation on the actin–myosin interaction. (**A**) Contact region between the cardiomyopathy (CM) loop of NM2C (blue) and residue E334 (orange) in cytoskeletal γ-actin (PDB ID: 5JLH) (*von der Ecken et al., 2016*). Actin interacts with a core contact triad in the myosin CM loop, which in the case of NM2C consists of lysine K429 and hydrophobic residues I420 and V427. (**B**) Multiple-sequence alignment of the CM loop of selected cytoskeletal myosin isoforms. The alignment shows complete sequence identity for NM2C, NM2A, and conservation of residues contributing to the core contact triad in class 5, class 19, and selected members of class 1 myosins (blue boxes). Residue numbering refers to the NM2C primary structure. (**C**) Mutation E334Q leads to a weakening of complementary electrostatic interactions. The interaction interface involving the CM loop of NM2C and the region affected by the mutation on cytoskeletal γ-actin is visualized by coulombic surface coloring. Positive potentials are shown in blue, and negative potentials in red.

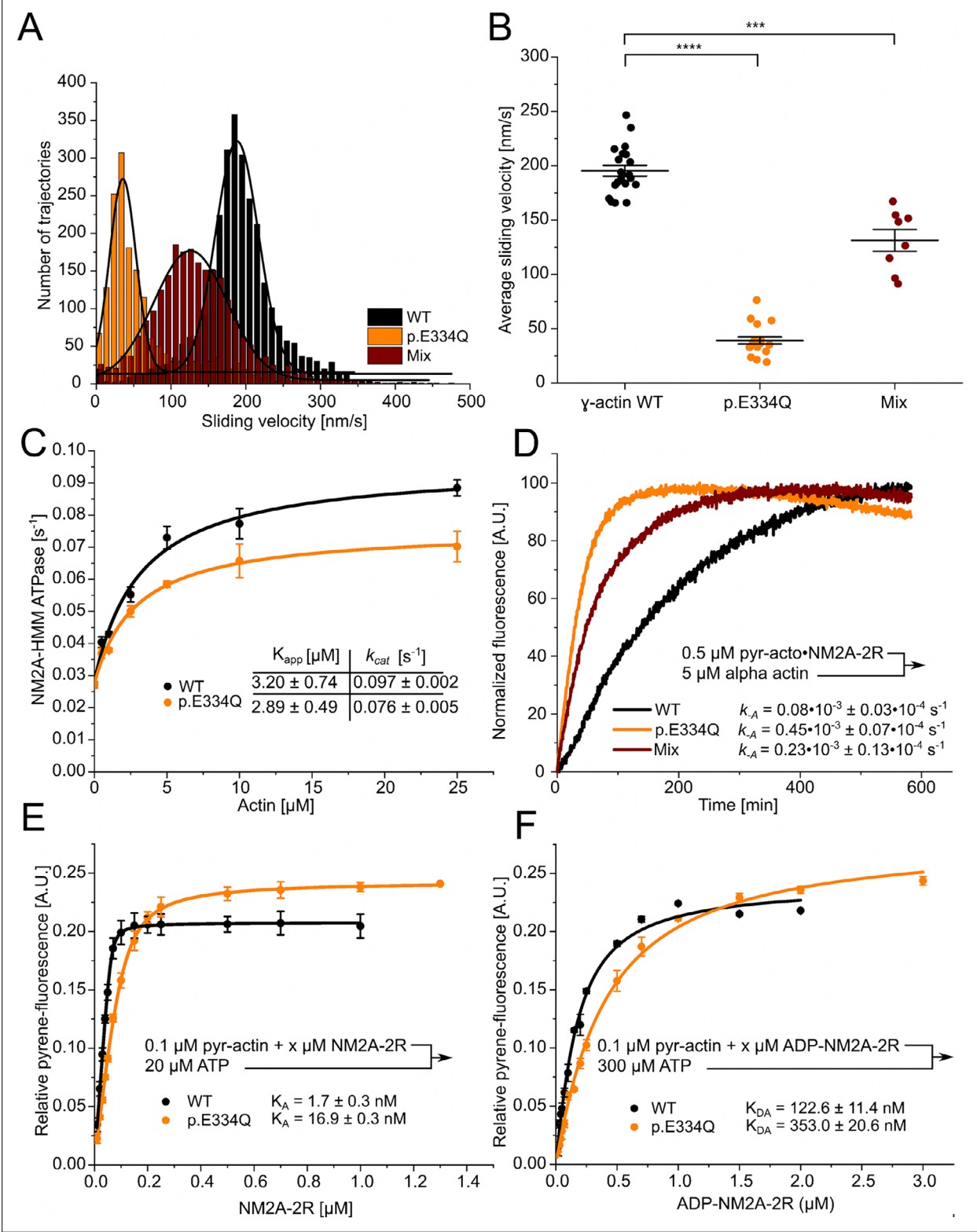

**Figure 8.** Analysis of the interaction of human NM2A with WT and p.E334Q γ-actin. (**A**) The interaction of surface-immobilized NM2A-HMM with WT γ-actin filaments, p.E334Q γ-actin filaments, and heterofilaments (1:1 mixture) was analyzed using the unloaded in vitro motility assay. Representative velocity distributions are obtained from recorded trajectories of WT and mutant filaments in a single experiment. The average sliding velocity of the filaments in each experiment was determined by applying a Gaussian fit (black line) to the obtained velocity distributions. (**B**) Secondary plot of all measured sliding velocities. Each data point represents a single experiment (N = 20 for WT filaments, N = 18 for p.E334Q filaments, N = 8 for

*Figure 8 continued on next page*

**eLife** Research article

Structural Biology and Molecular Biophysics

*Figure 8 continued*

heterofilaments) in which the sliding velocities of a minimum of 600 filaments have been analyzed. The average sliding velocity in each experiment was determined as described in (**A**). Data is shown as the mean ± SEM. (**C**) The ability of p.E334Q and WT filaments to activate the NM2A-HMM ATPase was analyzed using a NADH-coupled enzymatic assay. The data is best described by a hyperbolic fit that yields $K_{app}$ (apparent $K_m$ for actin) and $k_{cat}$ (maximal turnover). Data is shown as the mean ± SD of three individual experiments. (**D**) Determination of the dissociation rate constant ($k_{-A}$) for the interaction of NM2A-2R with p.E334Q or WT γ-actin in the absence of nucleotide. $k_A$ was determined in displacement experiments by chasing pyrene-labeled acto•NM2C-2R with a large excess of unlabeled F-actin. Shown are representative traces. The shown $k_A$ values are the results of 5–11 individual experiments. Data were fitted using a single exponential function. (**E, F**) The affinity of WT and p.E334Q γ-actin for NM2A-2R in the absence ($K_A$) and presence of saturating ADP concentrations ($K_{DA}$) was determined using the method developed by *Kurzawa and Geeves, 1996*. Increasing concentrations of myosin were dissociated from pyrene-labeled WT or p.E334Q F-actin by mixing the complex with excess ATP. The obtained fluorescence amplitudes of the individual transients were plotted against the corresponding myosin concentrations. A quadratic equation (see 'Materials and methods') was fitted to the data, which yields the respective affinity. Data is shown as the mean ± SD of three individual experiments.

The online version of this article includes the following video and figure supplement(s) for figure 8:

**Figure supplement 1.** Schematic representation of the basic reaction pathway of myosin- and actomyosin-catalyzed ATP hydrolysis and ATP conversion.

**Figure supplement 2.** Determination of the dissociation rate constants ($k_{-A}$) for the interaction of NM2C-2R with p.E334Q or WT γ-actin in the absence of nucleotide.

**Figure supplement 3.** Analysis of the apparent ATP-affinity ($1/K_1$), the maximal ATP-induced dissociation rate of NM2A from actin ($k_{+2}$), and the second-order rate constant for ATP binding ($K_1 k_{+2}$) for complexes formed between NM2A-2R and WT or p.E334Q filaments.

**Figure 8—video 1.** Sliding of WTγ-actin filaments (TRITC-phalloidin labeled) on surface-immobilized NM2A-HMM visualized by fluorescence microscopy.

https://elifesciences.org/articles/93013/figures#fig8video1

**Figure 8—video 2.** Sliding of p.E334Q γ-actin filaments (TRITC-phalloidin labeled) on surface-immobilized NM2A-HMM visualized by fluorescence microscopy.

https://elifesciences.org/articles/93013/figures#fig8video2

**Figure 8—video 3.** Sliding of mutant-WT heterofilaments (TRITC-phalloidin labeled) on surface-immobilized NM2A-HMM visualized by fluorescence microscopy.

https://elifesciences.org/articles/93013/figures#fig8video3

used. A similar change from $0.31×10^{-3}$ s$^{-1}$ to $1.70×10^{-3}$ s$^{-1}$ was observed for the analogous NM2C–2 R construct (*Figure 8—figure supplement 1*). Next, we determined the affinities of NM2A–2 R for WT and p.E334Q filaments in the absence of nucleotide ($K_A$) and the presence of saturating concentrations of ADP ($K_{DA}$). We performed affinity–titration experiments with pyrene–labeled, phalloidin–stabilized filaments and NM2A–2 R (*Kurzawa and Geeves, 1996*). The values measured under rigor conditions show a 9.9-fold reduced affinity of NM2A-2R for p.E334Q filaments ($K_A = 16.9 ± 1.3$ nM) compared to WT filaments ($K_A = 1.7 ± 0.3$ nM) (*Figure 8E*). When the experiment was performed in the presence of saturating concentrations of ADP, we obtained values for $K_{DA}$ of $353.0 ± 20.6$ nM in the presence of p.E334Q filaments and $122.6 ± 11.4$ nM with WT filaments (*Figure 8F*). The apparent ATP affinity ($1/K_1$), the maximal dissociation rate of NM2A from F-actin in the presence of ATP ($k_{+2}$), and the apparent second-order rate constant of ATP binding ($K_1 k_{+2}$) showed no significant differences for complexes formed between NM2A and WT or p.E334Q filaments (*Table 1*, *Figure 8—figure supplement 3*).

Next, we investigated the effects of the E334Q mutation on the motile activity of the processive motor Myo5A (*Figure 9*). Since the lysine residue corresponding to K429 in the CM loop of NM2C is also present in all human myosin-5 isoforms, the resulting changes in binding strength and thermodynamic coupling are expected to reduce the motor activity of Myo5A upon interactions with p.E334Q filaments. Structural studies have shown that the interaction between actin residue E334 and the CM loop in the Myo5A motor domain is maintained from the rigor to the ADP-bound state (*Pospich et al., 2021*). We performed unloaded in vitro motility assays to evaluate the effect of the mutation on the interaction with Myo5A-HMM. The rapid transition from weak to strong binding with a slow and rate-limiting ADP release leading to a high working ratio (*De La Cruz et al., 1999*) allows the in vitro motility assay to be performed on surfaces with a low motor density. In initial experiments, we found a Myo5A-HMM surface density of 770 molecules μm$^{-2}$ to be optimal for experiments with WT filaments. This much lower optimal motor surface density compared to the NM2A-HMM construct, which requires densities above 10,000 molecules per μm$^2$, is a manifestation of the processive properties of the Myo5A-HMM construct. Movement was steady and filament breaks were practically non-existent

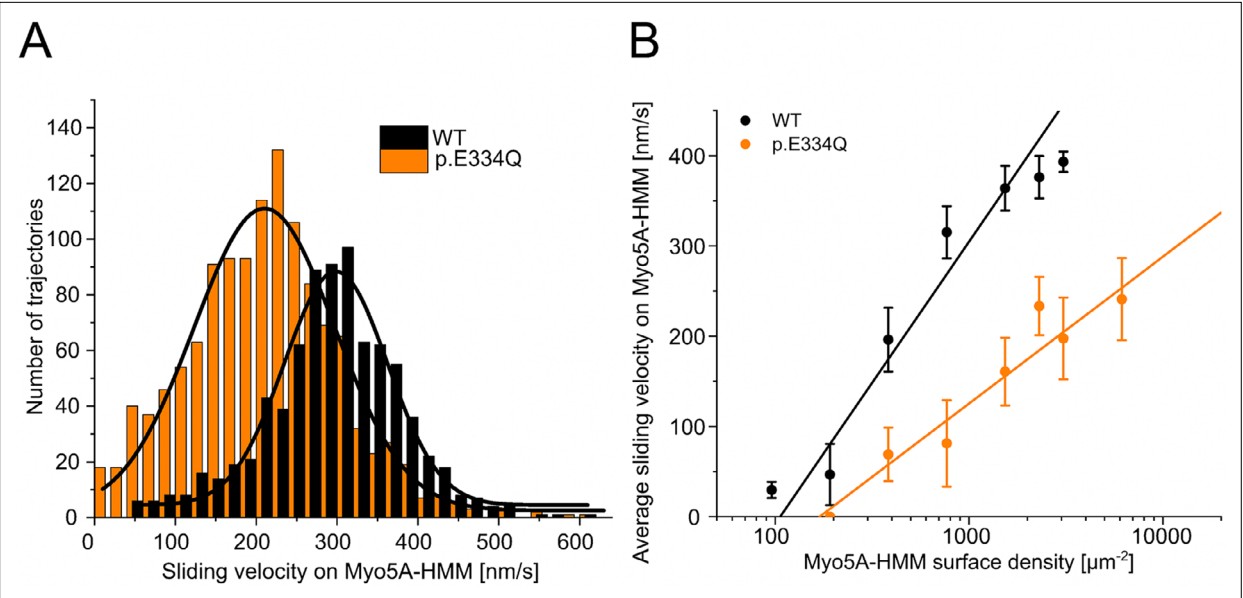

**Figure 9.** Analysis of the interaction of human Myo5A-HMM with WT and p.E334Q γ-actin in the unloaded in vitro motility assay at different Myo5A surface densities. (**A**) The interaction of surface-immobilized Myo5A-HMM with WT γ-actin and p.E334Q γ-actin filaments was analyzed using the unloaded in vitro motility assay. Representative velocity distributions were obtained from one individual experiment at the optimal Myo5A-HMM surface density for WT (770 μm⁻²) and p.E334Q filaments (3100 μm⁻²). The average sliding velocity of the filaments in each experiment was determined by applying a Gaussian fit (black line) to the obtained velocity distributions. (**B**) Semi-logarithmic plot summarizing all measured sliding velocities at different Myo5A-HMM surface densities. Data is shown as the mean ± SD of 3–10 individual experiments. The dependence of the average sliding velocity on surface motor density is best described in the semi-logarithmic graph by lines up to approximately 1000 motors μm⁻² for WT filaments and 7000 motors μm⁻² for p.E334Q filaments.

at 770 molecules μm⁻², while higher surface densities led to increased filament breaks, resulting in rapid fragmentation. At all Myo5A-HMM surface densities used, the sliding velocity was significantly reduced for p.E334Q compared to WT filaments. In experiments with p.E334Q filaments, a fourfold higher Myo5A-HMM surface density was required to achieve a similar actin surface decoration and optimal motile activity. Furthermore, filament breaks occurred less frequently, indicating a weaker actomyosin interaction. The number of moving filaments and their sliding velocity decreased sharply below surface densities of 770 molecules μm⁻² for WT filaments and 3100 molecules μm⁻² for p.E334Q filaments. Below a surface density of 380 Myo5A-HMM molecules μm⁻², no productive attachment events were observed with p.E334Q filaments. Thus, mutation E334Q in cytoskeletal γ-actin leads to an impairment of the interaction of the actin filament with Myo5A, similar to the changes observed with NM2A and NM2C, but with additional manifestations specific to processive motors.

## Discussion

At present, the exact pathogenesis of non-muscular actinopathies remains poorly understood. A rigorous molecular characterization of the contribution of disturbances in actin dynamics, contractility, and transport processes is necessary to develop an understanding of the diverse disease patterns.

In the past, the functional characterization of disease-associated mutations in cytoskeletal γ-actin focused on mutations p.T89I, p.K118N/M, p.E241K, p.T278I, p.P264L, p.P332A, and p.V370A, which all cause non-syndromic hearing loss. Two mutants showed abnormal polymerization behavior, indicated by a reduced polymerization rate (p.V370A) (*Bryan et al., 2006*) or an increased propensity for bundle formation (p.E241K) (*Morín et al., 2009*). The mutants p.K118N/M showed a perturbed interaction with the Arp2/3 complex, reducing branching efficiency (*Kruth and Rubenstein, 2012*; *Jepsen et al., 2016*). A significantly reduced protein stability was observed for two mutant proteins (p.P264L, p.P332A) (*Bryan et al., 2006*). Perturbed interactions with cofilin were observed for a range of mutations. p.T89I, p.E241K, p.T278I, and p.V370A are characterized by an increased sensitivity to cofilin-mediated filament severing (*Bryan and Rubenstein, 2009*). Interestingly, mutation p.P332A, close to

the analyzed mutation p.E334Q, enhances resistance toward cofilin-mediated severing (*Bryan and Rubenstein, 2009*). The perturbed cofilin interaction is the only overlap between the spectrum of molecular phenotypes observed for syndromic deafness-associated mutations and mutation p.E334Q, which results in a non-syndromic actinopathy. It has to be noted that all cited studies relied on the use of actin purified from budding yeast for their biochemical studies, which shows 91% sequence identity to human cytoskeletal γ-actin (*Ng and Abelson, 1980*).

Here, we describe mutation E334Q in cytoskeletal γ-actin that leads to a syndromic actinopathy with hypotonia and is not associated with non-syndromic hearing loss. Our biochemical analysis shows that the mutation does not affect the folding of the actin monomer or the stability of the monomer and the resulting filaments, demonstrating that mutations in the hinge region do not necessarily affect protein stability. The mutant protein displays a robust interaction with the VCA-activated Arp2/3 complex. Its polymerization kinetics are equivalent to those of the WT protein, with which it effectively copolymerizes in vitro. Based on these observations, we exclude functional haploinsufficiency as a potential disease mechanism and favor a mechanism in which p.E334Q integrates into the cytoskeletal actin networks of the cell but shows defects in the interaction with key binding partners. Consistent with this hypothesis and based on several structural studies (*Pospich et al., 2021*; *Behrmann et al., 2012*; *Robert-Paganin et al., 2021*; *Risi et al., 2021*; *Fujii and Namba, 2017*; *Gyimesi et al., 2020*; *Gong et al., 2022*; *Gurel et al., 2017*; *von der Ecken et al., 2016*) as well as our initial modeling of complexes formed by p.E334Q filaments, we observed perturbed interactions with cofilin-1 and human cytoskeletal myosin isoforms NM2A, NM2C, and Myo5A (*Table 1*).

We interpret the observed changes in the interaction with NM2A, NM2C, and Myo5A to primarily reflect the weakening of complementary interactions between actin residue E334 and a core contact triad in the myosin CM loop, which in the case of NM2C consists of K429 and hydrophobic residues I420 and V427 (*Figure 7*). Analogous changes in the interaction of mutant filaments with Myo1D, Myo1G, NM2B, Myo5B, Myo5C, and Myo19 are expected due to the presence of the core contact triad residues in their CM loops (*Figure 7B*). In contrast, the absence of a basic residue corresponding to K429 in NM2C and additional changes affecting their CM loop structure make it difficult to predict the consequences for the interaction of p.E334Q filaments with the cytoskeletal myosin isoforms Myo1A, Myo1C, Myo1E, Myo1F, Myo3A, Myo3B, Myo15, Myo18A, Myo18B, and Myo19.

In addition to primary consequences, reflected in the observed changes in the binding interactions with NM2A and cofilin-1, our results point to significant changes that result from allosteric perturbations. The results from our transient kinetic analysis of the complexes formed by p.E334Q with NM2A-2R indicate that reduced allosteric coupling between the actin- and nucleotide-binding sites within the myosin motor domain contributes to the observed decrease in motor activity. The coupling efficiency is given by the ratio of the dissociation equilibrium constants for actin binding in the absence ($K_A$) and presence of ADP ($K_{DA}$). The value for $K_{DA}/K_D$ obtained for the interaction of NM2A-2R in the presence of p.E334Q (20.9) is 3.5-fold lower than the value measured in the presence of WT actin (*Table 1*). Allosteric perturbations are also major contributors mediating the impact of mutation E334Q to the changes in elastic properties and the rate of actin filament disassembly in the presence of cofilin-1. Our results are consistent with the E334Q mutation slowing the initial complex formation due to less favorable charge interactions and allosteric perturbations having a major impact on subsequent steps.

Patients carrying mutations in *ACTB* or *ACTG1* predominately suffer from developmental disabilities, which can be directly linked to cortical malformations of varying severity. Therefore, neuronal cells and/or neuronal progenitor cells are assumed to be the most affected cell population in the patients (*Verloes et al., 2015*; *Di Donato et al., 2016*; *Rivière et al., 2012*). The actin composition of the neuron is thought to reach ratios between 1:1 and 1:2 for γ-actin to β-actin in the mature state after fluctuations during neuronal development (*Cheever and Ervasti, 2013*). Studies of the spatial localization of the cytoskeletal actin isoforms in the mature neuron are contradictory. A study showing that β- and γ-actin strongly colocalize in hippocampal and cerebral neurons (*Cheever et al., 2012*) appears to contradict previous studies, which reported spatial enrichment of β-actin in dynamic structures such as the dendritic spine and exclusion from less dynamic structures such as the axon shaft, where γ-actin was found to be enriched (*Weinberger et al., 1996*; *Micheva et al., 1998*; *Bassell et al., 1998*). Based on our results, we conclude that p.E334Q actin is readily produced and integrated into the cytoskeletal actin networks of neuronal cells, where it perturbs interactions with cytoskeletal

myosin isoforms that play essential roles in the function and maintenance of healthy neurons. The functions of myosin-5 isoforms in neuronal cells are all linked to their role as actin barbed-end-directed processive cargo transporters (*Hammer and Wagner, 2013*). Several of the myosin-5-driven transport processes are important contributors to synaptic plasticity, that is, the ability of the neurons to change synaptic strength based on previous activities (*Hammer and Wagner, 2013*).

The different NM2 isoforms are found in a wide range of cell types in the brain, but they show varying spatial expressions (*Murakami and Elzinga, 1992*; *Golomb et al., 2004*; *Rochlin et al., 1995*; *Kawamoto and Adelstein, 1991*). NM2A is predominately enriched in the vasculature of the brain, consistent with the importance of NM2A for angiogenesis (*Ma et al., 2020*). NM2B and NM2C are mainly found in growth cones and dendritic spines of neuronal cells (*Murakami and Elzinga, 1992*; *Golomb et al., 2004*; *Rochlin et al., 1995*). NM2 isoforms are known to contribute to various neuronal processes, but unlike for Myo5A, their exact role is less well understood. NM2B was shown to determine the maturation and morphology of the dendritic spines in hippocampal neurons and was additionally linked to basal synaptic function and plasticity (*Hodges et al., 2011*; *Ma et al., 2004*; *Rex et al., 2010*). NM2C and NM2B contribute to the differentiation of oligodendrocytes (*Domingues et al., 2020*). The NM2 isoforms compete with proteins of the ADF/cofilin family for binding to F-actin, and this competition is essential for stable actomyosin contractility and an intact cell cortex (*Wiggan et al., 2012*).

In line with this, ADF/cofilin family members are highly abundant in growth cones and dendritic spines and essential for spine morphology and determination of spine length by modulating actin turnover (*Star et al., 2002*; *Hotulainen et al., 2009*; *Namme et al., 2021*). Cofilin-mediated severing and nucleation were previously proposed to synergistically contribute to global actin turnover in cells (*Andrianantoandro and Pollard, 2006*; *Du and Frieden, 1998*). Our results show that the mutation affects these different cofilin functions in actin dynamics in opposite ways. Cofilin-mediated filament nucleation is more efficient for p.E334Q monomers, while cofilin-mediated severing of filaments containing p.E334Q is significantly reduced. The interaction of both actin monomers and actin filaments with ADF/cofilin proteins involves several distinct overlapping reactions. In the case of actin filaments, cofilin binding is followed by structural modification of the filament, severing and depolymerizing the filament (*De La Cruz and Sept, 2010*). Cofilin binding to monomeric actin is followed by the closure of the nucleotide cleft and the formation of stabilized 'long-pitch' actin dimers, which stimulate nucleation (*Andrianantoandro and Pollard, 2006*).

While the number of disease-associated mutations reported for *ACTB* and *ACTG1* is steadily increasing, the total number of patients affected remains low compared to patients with disease-associated mutations in muscle actin isoforms (*Parker et al., 2020*). High sequence and structure conservation across all actin isoforms enable us to extrapolate our findings obtained with cytoskeletal γ-actin to diseases associated with mutations of E334 in other actin isoforms. Mutations of residue E334 in α-skeletal actin (encoded by *ACTA1*) are reported to result in nemaline myopathy (E334K; *Laing et al., 2009*) or core myopathy (E334A; *Kaindl et al., 2004*; reviewed in *Parker et al., 2020*). The mutations either lead to the inversion of the charge at residue 334 (E334K) or the complete loss of charge (E334A). Our study shows that loss of charge at actin residue 334 leads to the loss of a conserved core contact between actin and the myosin motor domain of cytoskeletal class 2 myosin. Skeletal muscle myosin-2 is a central component of the sarcomere. The myosin forms the thick filaments that interact with the thin filaments formed by actin and regulatory proteins, generating force that powers muscle contraction (*Sweeney and Hammers, 2018*). Because the core contact triad that interacts with actin residue E334 is present in all class 2 myosin isoforms, it is reasonable to assume that mutations of residue E334 in α-skeletal actin affect the interaction with skeletal muscle myosin-2 to a similar extent, resulting in less efficient force generation and thereby contributing to the development of myopathy in patients.

Interestingly, a recent study analyzing the prevalence of cytoskeletal actin mutations in cancer using the cBioPortal database found that mutations in *ACTB* and *ACTG1* are associated with lymphoid cancers, specifically mutations in *ACTB* with diffuse large B-cell lymphoma and mutations in *ACTG1* with multiple myeloma. When the authors extended the analysis to all cancer types, they found that the observed mutations in the two genes were scattered across the entire actin sequence, with no apparent preference for a particular region. Nevertheless, the group identified residue E334 in *ACTB* as a potential hotspot as the missense variants p.E334Q/p.E334K are more frequent in all analyzed

datasets (*Witjes et al., 2020*). In this article, we investigated the impact of the E334Q exchange on the function of cytoskeletal γ-actin in the context of developmental diseases. However, the high sequence similarity between the cytoskeletal actin isoforms makes it highly likely that the E334Q mutation in cytoskeletal β-actin results in similar molecular perturbations of actin dynamics but with potentially different consequences for the organism due to varying enrichment of the two isoforms in functional compartments of the actin cytoskeleton. Although future studies are needed to determine the role of *ACTB* and *ACTG1* mutations as driver or passenger mutations in different cancers, our data show a possible link between impaired cytoskeletal actomyosin dynamics, cofilin-mediated actin turnover, and cancer development.

## Materials and methods

### Plasmids

The coding sequence of human cytoskeletal γ-actin (UniProt ID: P63261) was fused via a C-terminal linker to a His$_6$-tagged thymosin β4 moiety (UniProt ID: P62328), as previously described for yeast actin (UniProt ID: P60010) (*Noguchi et al., 2007*). The fusion protein was cloned into the multiple cloning site of the pFastBac-Dual vector under the control of the polyhedrin promotor. The plasmid encoding the mutant protein p.E334Q was generated by performing site-directed mutagenesis on the WT plasmid (forward primer: 5′-ATCGCACCCCCACAGCGCAAGTACTCG-3′, reverse primer: 5′-GATCTTGATCTTCATGGTGCTGGGCGC-3′). Motor domain-carrying fragments of human NM2A, NM2C, and Myo5A were produced using MYH9, MYH14, and MYO5A heavy chain sequences, respectively (UniProt IDs: P35579, Q7Z406, Q9Y4I1). For the production of HMM-like constructs of human NM2A and human Myo5A, expression vectors derived from pFastBac were used, as described previously (*Hundt et al., 2016*; *Reindl et al., 2022*). NM2A-HMM was co–produced with human light chains MYL6a (UniProt ID: P60660) and MYL12b (UniProt ID: O14950). Myo5A HMM was co-produced with human calmodulin (UniProt ID: P0DP23). NM2A-2R and NM2C-2R, single-headed, constitutively active motor domain constructs of human NM2A and NM2C-2C0 with an artificial lever arm region, were produced using pFastBac-derived expression vectors, as described previously (*Heissler and Manstein, 2011*; *Müller et al., 2013*). Human cofilin-1 (UniProt ID: P23528) was cloned into the expression vector pGEX-6P-2 (*Greve et al., 2022*). The human Arp2/3 complex consisting of subunits Arp2 (UniProt ID: P61160), Arp3 (UniProt ID: P61158), ArpC1B (UniProt ID: O15143), ArpC2 (UniProt ID: O15144), ArpC3A (UniProt ID: O15145), ArpC4 (UniProt ID: P59998), ArpC5 (UniProt ID: O15511), and a C-terminal FLAG tag on ArpC3A was produced using the expression vector pBIG2abc, provided by Roberto Dominguez (University of Pennsylvania, Philadelphia). The VCA domain of human N-WASP (UniProt ID: O00401) was cloned into pGEX-6P-2. Human capping protein (subunits α1/β2, UniProt ID: P52907/P47756) in pET-Duet-1 and eGFP-tagged human cofilin-1 (UniProt ID: P23528) in pGEX-6P-2 vector were provided by Jan Faix (Medizinische Hochschule Hannover, Hannover, Germany). Recombinant bacmids and viruses for the production of proteins in the Sf9 system were produced according to the *Bac-to-Bac* baculovirus expression system protocol (Thermo Fisher Scientific, Waltham, MA).

### Protein production and purification

We produced recombinant human cytoskeletal γ-actin and p.E334Q mutant protein as actin-thymosin β4-His$_6$ fusion protein using the baculovirus/*S. frugiperda* insect cell expression system. The C-terminal native actin residue is a Phe, which provides a cleavage site for chymotrypsin to remove downstream tag sequences completely. The tag sequence employed consists of a 14-amino acid linker (ASSGGSGSGGSGGA), 43 thymosin-β4 residues, followed by 6 histidine residues. To produce WT and mutant fusion proteins, $2 \times 10^9$ cells were harvested 72 hr after infection. Cells were pelleted, washed with PBS, shock frozen in liquid nitrogen, and stored at –80°C. Cell pellets from 2 l of Sf-9 culture were thawed for a typical preparation. For lysis, cells were resuspended in lysis buffer A (10 mM Tris pH 7.8, 5 mM CaCl$_2$, 10 mM imidazole, 1.25% Triton X-100, 100 mM KCl, 7 mM β-mercaptoethanol, 1 mM ATP, 1 mM PMSF, 100 µg/ mL TAME, 80 µg/mL TPCK, 2 µg/mL pepstatin, 5 µg/mL leupeptin) and sonicated. The lysate was cleared by centrifugation at 30,000 × *g* (30 min, 4°C). The cleared lysate was loaded onto 3 mL equilibrated PureCube 100 NiNTA agarose (Cube Biotech, Manheim am Rhein, Germany) and incubated for 2 h under constant rotation. Then, the material was washed extensively with 10 mM Tris pH 7.8, 0.2 mM CaCl$_2$, 10 mM imidazole, 50 mM KCl, 1 mM ATP, and

the actin-thymosin β4 fusion protein was eluted with 10 mM Tris pH 7.8, 250 mM imidazole, 0.2 mM CaCl$_2$, 1 mM ATP. Peak fractions were pooled and dialyzed overnight against G-buffer (10 mM Tris pH 7.8, 0.2 mM CaCl$_2$, 0.1 mM ATP, 0.5 mM DTT). The next day, protein aggregates were removed by centrifugation at 15,000 × g (20 min, 4°C). The supernatant was pooled and α-chymotrypsin (Sigma-Aldrich, Schnelldorf, Germany) was added at a w/w ratio of 1:300 (chymotrypsin:actin) to remove the tag sequences. The optimal digestion time varied from 45 to 80 min and needs to be determined for each batch of α-chymotrypsin. PMSF was added to a final concentration of 0.2 mM to stop the digestion. Actin was separated from the thymosin-His tag using polymerization–depolymerization steps. Unless otherwise stated, all protein purification steps were carried out at 4°C. F-actin was stored on ice. G-actin was concentrated to approximately 150 μM, flash frozen in liquid nitrogen, and stored at –80°C in G-buffer. Clarified G-actin, free of aggregates and oligomers, was always prepared fresh. G-actin stocks were thawed, diluted in G-buffer to the appropriate concentration, and centrifuged at 136,000 × g (15 min, 4°C). The top 2/3 of the centrifuged sample was carefully removed for use in further experiments. All myosin constructs were produced in the baculovirus/*S. frugiperda* insect cell expression system and purified as previously described for NM2A-HMM (*Greve et al., 2022*; *Hundt et al., 2016*; *Pathan-Chhatbar et al., 2018*), Myo5A-HMM (*Reindl et al., 2022*), the single-headed, constitutively active constructs NM2A-2R (*Heissler and Manstein, 2011*; *Müller et al., 2013*) and NM2C-2R (*Heissler and Manstein, 2011*; *Müller et al., 2013*). Production of the human Arp2/3 complex in insect cells followed the procedure described in *Zimmet et al., 2020*. Human cofilin-1 and human eGFP-cofilin-1 were produced as GST-fusion proteins in Rosetta2 (DE3) cells. Purification was carried out as previously described for human cofilin-1 (*Greve et al., 2022*).

Human GST-N-WASP-VCA was produced in ArcticExpress (DE3) cells. The cells were lysed by treatment with lysozyme followed by sonification in lysis buffer W (25 mM Tris pH 7.4, 50 mM NaCl, 1 mM EDTA, 1 mM DTT, 0.5% Triton X-100, 1 mM PMSF). The cleared cell lysate was loaded onto a self-packed GSH-Sepharose column. The column was then washed with five column volumes of wash buffer W (25 mM Tris pH 7.4, 50 mM NaCl, 1 mM EDTA, 1 mM DTT). GST-N-WASP-VCA was eluted from the column by applying 10 mM glutathione in wash buffer W. The eluted protein was concentrated and loaded onto a S75 16/600 size-exclusion chromatography column (GE Healthcare, Chicago). After chromatography, fractions containing pure protein were pooled, concentrated, and the concentration was determined via absorbance at 280 nm. The GST-N-WASP-VCA protein was frozen in liquid nitrogen and stored at –80°C.

Human capping protein (α1/β2) carrying a 6×His-tag was produced in Rosetta2 (DE3) cells. For purification, cells were lysed by lysozyme followed by sonification in lysis buffer CP (25 mM Tris pH 7.8, 250 mM NaCl, 1 mM EDTA, 1 mM PMSF, 1.5% Triton X-100, 10 mM imidazole, 7 mM β–mercaptoethanol). The cleared lysate was loaded onto PureCube 100 NiNTA agarose (Cube Biotech) in a gravity flow column. The agarose was washed with wash buffer CP (25 mM Tris pH 7.8, 250 mM NaCl, 30 mM imidazole) and the protein was then eluted with elution buffer CP (25 mM Tris pH 7.8, 50 mM NaCl, 250 mM imidazole). Fractions with pure capping protein were pooled, concentrated, and loaded onto a S200 16/600 size-exclusion chromatography column (GE Healthcare). The purified protein was frozen in liquid nitrogen and stored at –80°C.

α-skeletal actin (UniProt ID: P68139) was prepared from chicken pectoralis major muscle, as previously described for rabbit α-skeletal actin (*Lehrer and Kerwar, 1972*).

Unless otherwise stated, all purification steps were performed at 4°C.

## DNase I inhibition assay, DSF, and nucleotide exchange assay

The DNase I inhibition assay and DSF to study actin folding and thermal stability were performed as previously described (*Greve et al., 2022*).

Nucleotide exchange experiments were performed with freshly clarified G-actin as follows. The G-actin solution was passed through a Zeba spin desalting column (Thermo Fisher Scientific) equilibrated with G-buffer to remove excess nucleotide. The sample was then incubated with a twofold excess of ε-ATP or ε-ADP (Jena Bioscience, Jena, Germany) for 2 hr on ice. After incubation, excess nucleotide was removed by passing the sample through a Zeba spin desalting column equilibrated with 10 mM Tris pH 8.0 and 10 μM ε-ATP/ε-ADP. The concentration of the Ca$^{2+}$-ε-ATP/ε-ADP-G-actin solution was determined using the Coomassie protein assay (Thermo Fisher Scientific). Prior to the nucleotide exchange experiment, the Ca$^{2+}$-ε-ATP/ε-ADP-G-actin solution was supplemented with 10×

$Mg^{2+}$-exchange buffer (1 mM $MgCl_2$, 10 mM EGTA) to a final concentration of 0.1 mM $MgCl_2$ and 1 mM EGTA. The resulting solution was rapidly mixed 1:1 with assay buffer SF containing excess unlabeled ATP (10 mM Tris, pH 7.8, 0.2 mM $MgCl_2$, 0.2 mM ATP) using a HiTech Scientific SF61 stopped-flow system (TgK Scientific Limited, Bradford on Avon, UK). The ε-ATP/ε-ADP fluorescence was excited at 335 nm and monitored through a KV389 cutoff filter. The displacement of the fluorescent nucleotide analogs by ATP results in a decrease in the fluorescence amplitude, which is best described by a single exponential fit that yields the apparent rate of nucleotide dissociation.

## CD spectroscopy

Directly prior to the measurement, the freshly clarified G-actin was diluted to 0.05 mg/mL in 10 mM phosphate buffer pH 7.4. A buffer reference was prepared by mixing G-buffer with 10 mM phosphate buffer pH 7.4 in the same ratio. CD spectroscopy was performed with a PiStar 180 spectrometer (Applied Photophysics, Leatherhead, UK). Measurements were performed at 25°C using quartz cuvettes (0.3 cm path length) and nitrogen purging. CD spectra were recorded from 176 to 280 nm in steps of 1 nm using a bandwidth of 4 nm. For every sample, three spectra were recorded and subsequently averaged. The averaged buffer spectrum was subtracted, and the resulting protein spectrum smoothed using a Savitsky–Golay filter (*Savitzky and Golay, 1964*) in two consecutive iterations. The DichroWeb online platform was used to determine the secondary structure composition (*Whitmore and Wallace, 2004*). Data were analyzed based on the mean residue ellipticity $[\theta]_{MRE}$ of the sample using the CDSSTR algorithm and reference set 3 (optimized for 185–240 nm).

## Steady-state ATPase assay

The steady-state ATPase activity of F-actin was measured using a NADH-coupled enzymatic assay (*Furch et al., 1998*). Experiments were performed with 10 µM F-actin at 37°C in assay buffer A (25 mM HEPES pH 7.3, 5 mM $MgCl_2$, 50 mM KCl, 0.5 mM DTT) supplemented with the enzymatic system, which generates the quantifiable readout (1 mM ATP, 0.5 mM NADH, 0.5 mM 2-phosphoenolpyruvate, 0.02 mg/mL lactate dehydrogenase from rabbit muscle [Sigma-Aldrich], 0.05 mg/mL pyruvate kinase from rabbit muscle [Sigma-Aldrich], 0.5 mM NADH). For measurements of the basal and actin-activated NM2A-HMM ATPase, 0.5 µM MLCK-treated HMM was used. Phalloidin-stabilized WT or mutant F-actin was added over the range of 0–25 µM. The change in absorbance at 340 nm due to oxidation of NADH was recorded in a Multiskan FC Microplate Photometer (Thermo Fisher Scientific). The data were fitted to the Michaelis–Menten equation to obtain values for the actin concentration at half-maximal activation of ATP turnover ($K_{app}$) and for the maximum ATP turnover at saturated actin concentration ($k_{cat}$).

## TIRF microscopy-based assays

To distinguish between the impact of the mutation on filament nucleation and elongation, we performed TIRF microscopy-based assays using freshly clarified protein. Unlabeled WT or mutant G-actin was supplemented with 10% Atto 655-labeled WT or mutant G-actin for visualization of the filament assembly process. Glass coverslips were cleaned, surfaces chemically treated, and flow cells assembled as described previously (*Greve et al., 2022*). To induce polymerization, the G-actin solutions were diluted to a final concentration of 1 µM with 2× TIRF buffer (20 mM imidazole, 50 mM KCl, 1 mM $MgCl_2$, 1 mM EGTA, 0.2 mM ATP, 15 mM glucose, 20 mM β-ME, 0.25% methylcellulose, 0.1 mg/mL glucose oxidase, and 0.02 mg/mL catalase). The solutions were mixed, immediately flushed into flow cells, and image acquisition was started.

To follow filament disassembly, two-color TIRF experiments with Atto 655-labeled actin and eGFP-cofilin-1 were performed with preformed filaments that were 'aged' and capped at their barbed end by human capping protein. WT or mutant actin was polymerized for 3 hr at room temperature and then moved to 4°C overnight. 'Aged' F-actin and human capping protein were diluted into TIRF buffer to generate capped actin filaments. eGFP-cofilin-1 was added to final concentrations of 300 nM F-actin, 25 nM capping protein, and 100 nM eGFP-cofilin-1. Immediately adding eGFP-cofilin-1, the reaction mixture was flushed into the flow cell and image acquisition was started. Images were captured using a Nikon Eclipse TI-E inverted microscope equipped with a 100×/1.49 NA Apo oil immersion objective and Ixon3 897 EMCCD cameras (Andor, Belfast, UK) under control of the NIS software (Nikon, Minato, Japan). Images were taken every 3 s with an exposure time of 60ms.

To analyze actin polymerization in the presence of ABPs, the ABPs were first pre-diluted in KMEI buffer and diluted to their final concentration in TIRF buffer just before adding actin. Image acquisition was performed as described for experiments with actin only. Image analysis for all TIRF microscopy experiments was performed using ImageJ (*Rueden et al., 2017*). The elongation rate of actin filaments was determined by manually tracking individual filaments. Only filaments that could be tracked for at least four consecutive minutes were used to determine the elongation rate. Using these criteria, 20–30 filaments were tracked in each individual experiment (in a microscopic field of view of 81.41 × 81.41 µm). For quantification of two-color TIRFM experiments, 52 × 52 µm regions of interest from the entire microscopic field of view were used. The number of filaments observed throughout an experiment was determined using the *Analyze Particle* tool in ImageJ after background subtraction (sliding paraboloid, rolling ball radius: 25 pixels).

## Unloaded in vitro motility assay

The unloaded in vitro motility assay to study the interaction between WT and mutant actin and NM2A-HMM or Myo5A-HMM was essentially performed as previously described (*Greve et al., 2022*; *Reindl et al., 2022*; *Kron and Spudich, 1986*). Flow cells used in the in vitro motility assay were constructed using nitrocellulose-coated coverslips. NM2A-HMM requires phosphorylation of the associated regulatory light chain for full enzymatic activity. Purified NM2A-HMM was incubated with the GST-tagged kinase domain (residues 1425–1776) of human smooth muscle myosin light chain kinase (UniProt ID: A0A8I5KU53) at a molar ratio of 10:1 in 25 mM MOPS, pH 7.3, 50 mM KCl, 5 mM MgCl$_2$, 1 mM CaCl$_2$, 0.2 µM calmodulin, 3 µM regulatory light chain (MLC12b), 3 µM essential light chain (MLC6a), 1 mM DTT, and 1 mM ATP for 30 min at 30°C. In experiments with Myo5A-HMM, each buffer and actin solution was supplemented with 2 µM calmodulin to prevent calmodulin dissociation from Myo5A-HMM. Olympus IX70 or Olympus IX83 inverted fluorescence microscopes (Olympus, Hamburg, Germany) equipped with a 60×/1.49 NA PlanApo oil immersion objective and an Orca Flash 4.0 CMOS camera (Hamamatsu Photonics Deutschland GmbH, Herrsching, Germany) were used for image acquisition. All in vitro motility experiments were performed at a 37°C. Analysis of the recorded image series was performed in ImageJ (*Rueden et al., 2017*). The Plugin wrmTrck (*Nussbaum-Krammer et al., 2015*), developed by Jesper S. Pedersen, was used to determine the trajectories and corresponding velocities of the individual actin filaments. Only filaments that showed movement for at least 30 consecutive seconds were tracked and used to determine the average sliding velocity.

## Pyrene actin-based bulk polymerization assays

The polymerization and depolymerization of pyrene-labeled recombinant WT or mutant actin (5% pyrene-labeled WT or mutant actin) was monitored as a function of increasing or decreasing pyrene fluorescence using a Synergy 4 microplate reader (BioTek Instruments, Winooski, VT) and the built-in filter set (excitation: 340/30 nm; emission: 400/30 nm) at 25°C. For polymerization experiments, 20 µL of a 10 µM solution of pyrene-labeled Mg$^{2+}$-ATP-G-actin (WT or mutant) was placed in a black flat-bottom 96-well plate (BrandTech Scientific, USA). Then, 80 µL of 1.25× polymerization buffer (10 mM Tris, pH 7.8, 100 mM KCl, 5 mM MgCl$_2$, 0.5 mM EGTA, 0.1 mM DTT, and 0.1 mM ATP) was added to the wells using the built-in pipetting function of the plate reader, and the polymerization of actin was followed as an increase in pyrene fluorescence. For depolymerization experiments, 3 µL of a 20 µM F-actin solution was placed in one well of the plate. The solution was rapidly diluted by the addition of 297 µL G-buffer. The depolymerization of the filaments was monitored as a decrease in pyrene fluorescence.

The effects of Arp2/3 (+GST-N-WASP-VCA) and cofilin-1 on filament assembly or disassembly were analyzed by dispensing the actin solution and the ABP solution into the well by hand without mixing. Polymerization buffer or G-buffer was added, the reaction components were thereby mixed, and recording of the ensuing changes in fluorescence intensity was started.

The bulk polymerization rate was determined according to *Doolittle et al., 2013*. A linear regression was fitted to the linear region around the time point of half-maximal fluorescence. The depolymerization experiments were evaluated by fitting a single exponential function to the data, including a linear component that accounts for bleaching.

## Co-sedimentation experiments

Actin was polymerized at 20 µM for 3 hr at room temperature before being moved to 4°C overnight. For co-sedimentation experiments at pH 7.8, 5 µM of 'aged' F-actin and varying concentrations of human cofilin-1 were mixed in assay buffer C (10 mM Tris, pH 7.8, 100 mM KCl, 5 mM MgCl$_2$, 0.5 mM EGTA, 0.1 mM DTT, and 0.1 mM ATP) to a final volume of 50 µL in 1.5 mL polypropylene centrifuge tubes (Beckman Coulter, Brea, CA) suited for high-speed centrifugation. For experiments performed at pH 6.5, assay buffer C was supplemented with piperazine-*N,N'-bis*(2-ethanesulfonic acid) (PIPES, pH 6.5) to a final concentration of 100 mM to induce a pH shift from 7.8 to 6.5. Actin and cofilin-1 were added after the pH shift. Samples were incubated for 10 min at 23°C and then centrifuged at 136,000 × *g* (15 min, 4°C). The pellet and supernatant fractions were subjected to SDS-PAGE and the gel was imaged at a ChemiDoc-MP-gel documentation system (Bio-Rad Laboratories, Hercules, CA). The amount of actin and cofilin-1 in the pellet and supernatant was determined via densitometry using the Image Lab software (Bio-Rad Laboratories).

## Transient kinetic analysis of the actomyosin complex

Myosin binding to pyrene-labeled actin leads to strong pyrene fluorescence quenching. Mixing of the rigor-pyrene-acto-NM2A-2R/NM2C-2R complex with an excess of unlabeled actin allows the determination of $k_{-A}$ (dissociation rate of myosin from F-actin in the absence of nucleotide). Prior to the experiment, pyrene-labeled phalloidin-stabilized WT or mutant F-actin and the myosin solutions were treated with apyrase (Merck KGaA, Darmstadt, Germany) to remove residual nucleotide. Pyrene-labeled actin was mixed with a 1.5-fold excess of NM2A-2R/NM2C-2R in assay buffer P (20 mM MOPS pH 7.0, 100 mM KCl, 5 mM MgCl$_2$, 5 mM DTT) and incubated for 5 min on ice. The solution was mixed with an excess of chicken α-skeletal actin in assay buffer P to final concentrations of 0.5 µM pyrene-labeled actin, 0.75 µM NM2A-2R, and 5 µM α-skeletal actin. The solution was immediately transferred into the well of a black 96-well microplate with a transparent bottom and a lid (Corning, Corning, NY). Fluorescence was excited at 365 nm and measured at 407 nm in a CLARIOstar Plus microplate reader (BMG LABTECH, Offenburg, Germany) using bottom optics over 9 hr. The dissociation rate ($k_{-A}$) of myosin from actin was determined by applying a single exponential fit to the data.

The apparent affinity of NM2A-2R to WT or mutant F-actin in the absence ($K_A$) or presence ($K_{DA}$) of nucleotide was determined using the method developed by *Kurzawa and Geeves, 1996*. To determine $K_A$, all protein samples were treated with apyrase before the experiment. Then, 100 nM pyrene-labeled phalloidin-stabilized F-actin was incubated with increasing concentrations of NM2A-2R (10–1300 nM) for 5 min on ice in assay buffer P without DTT. The solutions were shot against 20 µM ATP at a HiTech Scientific SF61 stopped-flow system (TgK Scientific Limited) in a 1:1 ratio at 20°C. Fluorescence was excited at 365 nm and recorded using a KV389 cutoff filter. At least five transients were averaged for each concentration of NM2A-2R and fitted using a single exponential function. The relative fluorescence amplitudes were plotted against the pre-mix myosin concentrations. The affinity was determined by applying a quadratic equation to the dataset, where $A$ is the amplitude, $A_{max}$ the amplitude under saturation, [F-actin] the actin concentration prior mix, [NM2A-2R] the variable NM2A-2R concentration, and $K$ the dissociation constant in the absence or presence of nucleotide.

$$A = A_{max} \times \left( [F-Actin] + [NM2A-2R] + K \right) - \sqrt{\frac{\left( [F-Actin] + [NM2A-2R] + K \right)^2 - 4 \times [F-Actin]}{2 \times [F-Actin]}}$$

Slight modifications of the experimental protocol were necessary to determine $K_{DA}$. Incubation with apyrase was omitted; instead, the pyrene actomyosin complex was incubated with 30 µM ADP (Merck KGaA) for 20 min on ice prior to measurement. To account for the lower affinity in the presence of ADP, the range of used myosin concentrations was extended up to 3000 nM. Additionally, the concentration of ATP in the mixing experiment was increased to 300 µM pre-mix. Data was recorded and evaluated as done for experiments without nucleotide.

The apparent ATP affinity of the actomyosin complex was determined by mixing the apyrase-treated, pyrene-labeled, phalloidin-stabilized actomyosin complex with increasing concentrations of ATP at the stopped-flow system. Fitting an exponential function to the individual transients yields the ATP-dependent dissociation rate of NM2A-2R from F-actin ($k_{obs}$). The $k_{obs}$ values were plotted against the corresponding ATP concentrations and a hyperbola was fitted to the data. The fit yields

the apparent ATP affinity (1 /$K_1$) of the actomyosin complex and the maximal dissociation rate $k_{+2}$. The apparent second-order rate constant for ATP binding ($K_1k_{+2}$) was determined by applying a linear fit to the data obtained at low ATP concentrations (0–25 µM).

## Molecular modeling

Homology models of human cytoskeletal γ-actin, human cofilin-1, and the human NM2A motor domain were generated with MODELLER (*Sali and Blundell, 1993*). Models of filamentous human cytoskeletal γ-actin with bound cofilin-1 or NM2A motor domain were generated based on cryo-EM structures obtained from the Protein Data Bank of the barbed end side of a cofilactin cluster (PDB: 6UC4, 6UBY, 6VAO) (*Huehn et al., 2020*) and a human cytoplasmic actomyosin complex (PDB: 5JLH) (*von der Ecken et al., 2016*). The structures were accessed on May 25, 2023. Mutation E334Q was introduced using the mutagenesis tool in PyMOL (*Schrödinger, 2015*). PyMOL and ChimeraX (*Goddard et al., 2018*; *Pettersen et al., 2021*) were used for distance measurements and visualization of Coulomb surfaces.

Identification and alignment of the CM loop region of cytoskeletal myosin isoforms were performed using the match function of ChimeraX by superimposing and aligning cytoskeletal myosin isoform structures on NM2C (pdb-NM2C: 5JLH; pdb-Myo5A: 7PM5; pdb-Myo5C: 5HMP; pdb-Myo1D: AF-O94832-F1; pdb-Myo15: 7UDT; pdb-Myo19: AF-Q96H55-F1). Where appropriate, the corresponding human sequences were retrieved from UniProt and structurally aligned.

Normal mode analyses (NMA) were performed using the Elastic Network Model (ENM) elNémo (*Suhre and Sanejouand, 2004*) and an nrbl value of 5.

## Data analysis

Data analysis and graph plotting were performed with Origin 2023 (OriginLab Corporation. MA). Errors are given as standard deviation (SD) based on three independent experiments if not otherwise specified. The significance of the data was evaluated in Origin 2023 using a two-sample *t*-test (ns = p>0.05, *p≤0.05, **p≤0.01, ***p≤0.001, ****p≤0.0001).

## Materials availability

Unique reagents generated in this study are available from the lead contact with a completed Materials Transfer Agreement.

## Acknowledgements

We thank Roberto Dominguez (University of Pennsylvania, Philadelphia, USA) for providing vector pBIGBac-Arp2/3 and Jan Faix (Medizinische Hochschule Hannover, Hannover, Germany) for providing vectors pGEX-eGFP-cofilin-1 and pETDuet-CP. DJM was supported by grants from Deutsche Forschungsgemeischaft (DFG) (MA1081/23-1, MA1081/28-1), JNG, AM, NDD, and DJM are members of the European Union's Horizon 2020 research and innovation program under the EJP RD COFUND-EJP No. 825575 with support from the German Federal Ministry of Education and Research under Grant Agreement 01GM1922B. DJM is a member of the Cluster of Excellence RESIST (EXC 2155; DFG-Project ID: 39087428-B11). JNG is supported by the HiLF I grant for early career researcher from Medizinische Hochschule Hannover. We gratefully acknowledge support provided by the Core Unit for Structural Biochemistry, the Core Unit for Laser Microscopy at MHH. Computing time was provided on supercomputers Lise and Emmy at NHR@ZIB and NHR@Göttingen, as part of the Alliance for National High Performance Computing (NHR) infrastructure. The calculations for this research were conducted with computing resources under the project ID nib00018. TR was enrolled in the PhD program Molecular Medicine of Hannover Biomedical Research School (HBRS).

## Additional information

### Funding

| Funder | Grant reference number | Author |
| --- | --- | --- |
| Deutsche Forschungsgemeinschaft | MA1081/23-1 | Dietmar J Manstein |
| Deutsche Forschungsgemeinschaft | MA1081/28-1 | Dietmar J Manstein |
| Bundesministerium für Bildung und Forschung | 01GM1922B | Dietmar J Manstein |
| Horizon 2020 | EJP RD COFUND 825575 | Nataliya Di Donato |
| Hannover Medical School | HiLF I | Johannes N Greve |
| Norddeutscher Verbund für Hoch- und Höchstleistungsrechnen | nib00018 | Dietmar J Manstein |

The funders had no role in study design, data collection and interpretation, or the decision to submit the work for publication.

### Author contributions

Johannes N Greve, Conceptualization, Data curation, Formal analysis, Validation, Investigation, Visualization, Methodology, Writing – original draft, Writing – review and editing; Anja Marquardt, Robin Heiringhoff, Formal analysis, Investigation, Visualization, Methodology; Theresia Reindl, Claudia Thiel, Resources, Investigation; Nataliya Di Donato, Conceptualization, Resources, Funding acquisition, Writing – review and editing; Manuel H Taft, Resources, Validation; Dietmar J Manstein, Conceptualization, Resources, Data curation, Formal analysis, Supervision, Funding acquisition, Validation, Methodology, Project administration, Writing – review and editing

### Author ORCIDs

Theresia Reindl ⓘ https://orcid.org/0000-0001-9633-1899
Manuel H Taft ⓘ http://orcid.org/0000-0001-5853-8629
Dietmar J Manstein ⓘ https://orcid.org/0000-0003-0763-0147

Reviewer #1 (Public Review): https://doi.org/10.7554/eLife.93013.3.sa1
Reviewer #2 (Public Review): https://doi.org/10.7554/eLife.93013.3.sa2
Author Response https://doi.org/10.7554/eLife.93013.3.sa3

---

## Additional files

### Supplementary files
• MDAR checklist

### Data availability

Figure 1—figure supplement 1—source data 1, Figure 1—figure supplement 1—source data 2, Figure 1—figure supplement 1—source data 3 and Figure 5—figure supplement 2—source data 1 contain SDS-gels used to generate Figure 1—figure supplement 1 and Figure 5—figure supplement 2.

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
