## [Editor Report · eLife assessment]

This study presents a **valuable** characterization of the biochemical consequences of a disease-associated point mutation in a nonmuscle actin. The study uses well-characterized in vitro assays to explore function. The data are **convincing** and should be helpful to others.

---

## [Referee Report · Reviewer #1 (Public Review)]

This paper is of importance to scientists interested in molecular mechanisms by which actin point mutations affect its function to ultimately lead to disease states. This work thoroughly characterizes the effect of the E334Q mutation in cytoplasmic gamma-actin on two binding partners: cofilin and myosin (non-muscle myosin 2 and myosin 5). Overall, the data showing effects on cofilin function and myosin binding are convincing and the experiments performed expertly using state-of-the art approaches. Additional binding partners of actin that were not examined here may also have altered function when interacting with the mutant actin.

Comments on revised version:

The authors seem to have done a pretty thorough job with the rebuttal.

---

## [Referee Report · Reviewer #2 (Public Review)]

Greve et al. investigated the effects of a disease associated gamma-actin mutation (E334Q) on actin filament polymerization, association of selected actin-binding proteins, and myosin activity. Recombinant wildtype and mutant proteins expressed in sf9 cells were found to be folded and stable, and the presence of the mutation altered a number of activities. Given the location of the mutation, it is not surprising that there are changes in polymerization and interactions with actin binding proteins.

Comments on revised version:

I have nothing to add and am satisfied with the rebuttal.

---

## [Author Response]

The following is the authors’ response to the original reviews.

**eLife assessment**
This study presents a useful characterization of the biochemical consequences of a disease-associated point mutation in a nonmuscle actin. The study uses solid and well-characterized in vitro assays to explore function. In some cases the statistical analyses are inadequate and several important in vitro assays are not employed.
**Public Reviews:**

**Reviewer #1 (Public Review):**
Strengths:The authors first perform several important controls to show that the expressed mutant actin is properly folded, and then show that the Arp2/3 complex behaves similarly with WT and mutant actin via a TIRF microscopy assay as well as a bulk pyrene-actin assay. A TIRF assay showed a small but significant reduction in the rate of elongation of the mutant actin suggesting only a mild polymerization defect.Based on in silico analysis of the close location of the actin point mutation and bound cofilin, cofilin was chosen for further investigation. Faster de novo nucleation by cofilin was observed with mutant actin. In contrast, the mutant actin was more slowly severed. Both effects favor the retention of filamentous mutant actin. In solution, the effect of cofilin concentration and pH was assessed for both WT and mutant actin filaments, with a more limited repertoire of conditions in a TIRF assay that directly showed slower severing of mutant actin.Lastly, the mutated residue in actin is predicted to interact with the cardiomyopathy loop in myosin and thus a standard in vitro motility assay with immobilized motors was used to show that non-muscle myosin 2A moved mutant actin more slowly, explained in part by a reduced affinity for the filament deduced from transient kinetic assays. By the same motility assay, myosin 5A also showed impaired interaction with the mutant filaments.The Discussion is interesting and concludes that the mutant actin will co-exist with WT actin in filaments, and will contribute to altered actin dynamics and poor interaction with relevant myosin motors in the cellular context. While not an exhaustive list of possible defects, this is a solid start to understanding how this mutation might trigger a disease phenotype.

We thank the reviewer for the positive evaluation of our work.

Weaknesses:Potential assembly defects of the mutant actin could be more thoroughly investigated if the same experiment shown in Fig. 2 was repeated as a function of actin concentration, which would allow the rate of disassembly and the critical concentration to also be determined.

The polymerization rate of individual filaments observed in TIRFM experiments showed only minor changes, as did the bulk-polymerization rate of 2 µM actin in pyrene-actin based experiments. Therefore, we decided not to perform additional pyrene-actin based experiments, in which we titrate the actin concentration, as we expect only very small changes to the critical concentration. Instead, we focused on the disturbed interaction with ABPs, as we assume these defects to be more relevant in an in vivo context. Using pyrene-based bulkexperiments, we did determine the rate of dilution-induced depolymerization of mutant filaments and compare them with the values determined for wt (Figure 5A, Table 1).

The more direct TIRF assay for cofilin severing was only performed at high cofilin concentration (100 nM). Lower concentrations of cofilin would also be informative, as well as directly examining by the TIRF assay the effect of cofilin on filaments composed of a 50:50 mixture of WT:mutant actin, the more relevant case for the cell.

The TIRF assay for cofilin severing was performed initially over the cofilin concentration range from 20 to 250 nM. The results obtained in the presence of 100 nM cofilin allow a particularly informative depiction of the differences observed with mutant and WT actin. This applies to the image series showing the changes in filament length, cofilin clusters, and filament number as well as to the graphs showing time dependent changes in the number of filaments and total actin fluorescence. We have not included the results for a 50:50 mixture of WT:mutant actin because its attenuating effect is documented in several other experiments in the manuscript.

The more appropriate assay to determine the effect of the actin point mutation on class 5 myosin would be the inverted assay where myosin walks along single actin filaments adhered to a coverslip. This would allow an evaluation of class 5 myosin processivity on WT versus mutant actin that more closely reflects how Myo5 acts in cells, instead of the ensemble assay used appropriately for myosin 2.

Our results with Myo5A show a less productive interaction with mutant actin filaments as indicated by a 1.7-fold reduction in the average sliding velocity and an increase in the optimal Myo5A-HMM surface density from 770 to 3100 molecules per µm2. These results indicate a reduction in binding affinity and coupling efficiency, with a likely impact on processivity. We expect only a small incremental gain in knowledge about the extent of changes by performing additional experiments with an inverted assay geometry, given that under physiological conditions the motor properties of Myo5A and other cytoskeletal myosins are modulated by other factors such as the presence of tropomyosin isoforms and other actin binding proteins.

**Reviewer #2 (Public Review):**
Greve et al. investigated the effects of a disease-associated gamma-actin mutation (E334Q) on actin filament polymerization, association of selected actin-binding proteins, and myosin activity. Recombinant wildtype and mutant proteins expressed in sf9 cells were found to be folded and stable, and the presence of the mutation altered a number of activities. Given the location of the mutation, it is not surprising that there are changes in polymerization and interactions with actin binding proteins. Nevertheless, it is important to quantify the effects of the mutation to better understand disease etiology.

We thank the reviewer for the positive evaluation of our work.

Some weaknesses were identified in the paper as discussed below.Throughout the paper, the authors report average values and the standard-error-of-the-mean (SEM) for groups of three experiments. Reporting the SEM is not appropriate or useful for so few points, as it does not reflect the distribution of the data points. When only three points are available, it would be better to just show the three different points. Otherwise, plot the average and the range of the three points.

We have gone through the manuscript carefully to correct any errors in the statistics, as explained below.

Figure 1B, 5B, 5C, 5D, 8D, 9B, and 8 – figure supplement 2 all show the mean ± SD, as also correctly reported for Figure 8E and 8F in the figure legend. The statement, that these figures show the mean ± SEM was inaccurate. We corrected this mistake for all the listed figures. Furthermore, we now give the exact N for every experiment in the figure legend.

Figure 2C, 2E, 2F, 4B, 5A, 6B-E showed the mean ± SEM. As suggested by the reviewer, we corrected the figures to show the mean ± SD.

We still refer to the mean ± SEM in Figure 2B, where elongation rates for more than 100 filaments were recorded, and in Figure 8B, where sliding velocities for several thousand actin filaments were measured.

The description and characterization of the recombinant actin is incomplete. Please show gels of purified proteins. This is especially important with this preparation since the chymotrypsin step could result in internally cleaved proteins and altered properties, as shown by Ceron et al (2022). The authors should also comment on N-terminal acetylation of actin.

We added an additional figure showing the purification strategy for the recombinant cytoskeletal γ –actin WT and p.E334Q protein with exemplary SDS-gels from different stages of purification (Figure 1 – figure supplement 1).

In a previous paper, we reported the mass spectrometric analysis of the post-translational modifications of recombinant human β- and γ-cytoskeletal actin produced in Sf-9 cells. (Müller et al., 2013, Plos One). Recombinant actin showing complete N-terminal processing resulting in cleavage of the initial methionine and acetylation of the following aspartate (β-actin) or glutamate (γ-actin) is the predominant species in the analyzed preparations (> 95 %). While the recombinant actin in the 2013 study was produced tag-free and purified by affinity chromatography using the column-immobilized actin-binding domain of gelsolin (G4-G6), we have no reason to assume that the purification strategy using the actin-thymosin-β4 changes the efficiency of the N-terminal processing in Sf-9 cells. This is supported by our, yet unpublished, mass-spectrometric studies on recombinant human α-cardiac actin purified using the actin- thymosin-β4 fusion construct, which revealed actin species with an acetylated aspartate-3. This N-terminal modification of α-cardiac actin is catalyzed by the same actinspecific acetyltransferase (NAA80) as the acetylation of asparate-2 or glutamate-2 in cytoskeletal actin isoforms (Varland et al., 2019, Trends in Biochemical Sciences). Furthermore, additional studies that used the actin-thymosin-β4 fusion construct for the production of recombinant human cytoskeletal actin isoforms in Pichia pastoris reported robust N-terminal acetylation, when the actin was co-produced with NAA80 (In contrast to Sf-9 cells, NAA80 is not endogenously expressed in Pichia pastoris) (Hatano et al., 2020, Journal of Cell Science).

We therefore, added the following statement to the manuscript:

“Purification of the fusion protein by immobilized metal affinity chromatography, followed by chymotrypsin–mediated cleavage of C–terminal linker and tag sequences, results in homogeneous protein without non–native residues and native N-terminal processing, which includes cleavage of the initial methionine and acetylation of the following glutamate. “

The authors do not use the best technique to assess actin polymerization parameters. Although the TIRF assay is excellent for some measurements, it is not as good as the standard pyrene-actin assays that provide critical concentration, nucleation, and polymerization parameters. The authors use pyrene-actin in other parts of the paper, so it is not clear why they don't do the assays that are the standard in the actin field.

The polymerization rate of individual filaments observed in TIRFM experiments showed only minor changes, as did the bulk-polymerization rate of 2 µM actin in pyrene-actin based experiments. Therefore, we decided not to perform additional pyrene-actin based experiments, in which we titrate the actin concentration, as we expect only very small changes to the critical concentration. Instead, we focused on the disturbed interaction with ABPs, as we assume these defects to be more relevant in an in vivo context. Using pyrene-based bulkexperiments, we did determine the rate of dilution-induced depolymerization of mutant filaments and compare them with the values determined for WT (Figure 5A, Table 1).

The authors' data suggest that, while the binding of cofilin-1 to both the WT and mutant actins remains similar, the major defect of the E334Q actin is that it is not as readily severed/disassembled by cofilin. What is missing is a direct measurement of the severing rate (number of breaks per second) as measured in TIRF.

The severing rate as measured in TIRF is dependent on a number of parameters in a nonlinear manner. Therefore, we opted to show the combination of images directly showing the progress of the reaction and graphs summarizing the concomitant changes in cofilin clusters, actin filaments, actin-related fluorescence intensity and cofilin-related fluorescence intensity.

Figure 4 shows that the E334Q mutation increases rather than decreases the number of filaments that spontaneously assemble in the TIRF assay, but it is unclear how reduced severing would lead to increased filament numbers, rather, the opposite would be expected. A more straightforward approach would be to perform experiments where severing leads to more nuclei and therefore enhances the net bulk assembly rate.

Figure 4 shows polymerization experiments that were started from ATP-G-actin in the presence of cofilin-1. These experiments show clearly that, especially at the higher cofilin-1 concentration (100 nM), the filament number is strongly increased in experiments performed with mutant actin. Inspection of the corresponding videos of these TIRFM experiments suggest that the increased number of filaments must result from an increased number of de novo nucleation events and not primarily from a mutation-induced change in severing susceptibility. The observation of a cofilin-stimulated increase in the de novo nucleation efficiency of actin was initially described by Andrianantoandro & Pollard (2006, Molecular Cell) using TIRFMbased experiments and is thought to arise from the stabilization of thermodynamically unfavorable actin dimers and trimers by cofilin. While the exact role of this cofilin-mediated effect in vivo is not completely clear, it is thought to contribute to cofilin-meditated actin dynamics synergistically with cofilin-mediated severing. It is therefore necessary, to clearly distinguish between the two effects of cofilin in vitro: stimulation of de novo nucleation and stimulation of filament disassembly. Our data indicated that the E334Q mutation affects these two effects differentially, as we state in the abstract and in the discussion.

Abstract: “E334Q differentially affects cofilin-mediated actin dynamics by increasing the rate of cofilin-mediated de novo nucleation of actin filaments and decreasing the efficiency of cofilin-mediated filament severing.”

Discussion: “Cofilin-mediated severing and nucleation were previously proposed to synergistically contribute to global actin turnover in cells (Andrianantoandro & Pollard, 2006; Du & Frieden, 1998). Our results show that the mutation affects these different cofilin functions in actin dynamics in opposite ways. Cofilin-mediated filament nucleation is more efficient for p.E334Q monomers, while cofilin-mediated severing of filaments containing p.E334Q is significantly reduced. The interaction of both actin monomers and actin filaments with ADF/cofilin proteins involves several distinct overlapping reactions. In the case of actin filaments, cofilin binding is followed by structural modification of the filament, severing and depolymerizing the filament (De La Cruz & Sept, 2010). Cofilin binding to monomeric actin is followed by the closure of the nucleotide cleft and the formation of stabilized “long-pitch” actin dimers, which stimulate nucleation (Andrianantoandro & Pollard, 2006)”.

We interpret the reviewer's suggestion to mean that additional pyrene-actin-based bulk polymerization experiments should be performed to investigate the bulk-polymerization rate of ATP-G-actin in the presence of cofilin-1. In our understanding, these experiment would not provide additional value as (1) An observed increase of the bulk-polymerization rate cannot be directly correlated to a change of the efficiency of de novo nucleation or severing and (2) the effect of the mutation on cofilin-mediated filament disassembly was extensively analyzed in other experiments starting from preformed actin filaments. Moreover, our results are consistent with in silico modelling and normal mode analysis of the WT and mutant actin-cofilin complex.

Figure 5 A: in the pyrene disassembly assay, where actin is diluted below its critical concentration, cofilin enhances the rate of depolymerization by generating more free ends. The E334Q mutation leads to decreased cofilin-induced severing and therefore lower depolymerization. While these data seem convincing, it would be better to present them as an XY plot and fit the data to lines for comparison of the slopes.

We now present the data as suggested by the reviewer. Furthermore, we determined the apparent second-order rate constant for cofilin-induced F-actin depolymerization (kc) to quantify the observed differences between WT, mutant and heterofilaments, as suggested by the reviewer.

The paragraph describing these results was changed accordingly:

“The observed rate constant values are linearly dependent on the concentration of cofilin–1 in the range 0–40 nM, with the slope corresponding to the apparent second– order rate constant (kC) for the cofilin-1 induced depolymerization of F–actin. In experiments performed with p.E334Q filaments, the value obtained for kC was 4.2-fold lower (0.81 × 10-4 ± 0.08 × 10-4 nM-1 s-1) compared to experiments with WT filaments (3.42 × 10-4 ± 0.22 × 10-4 nM-1 s-1). When heterofilaments were used, the effect of the mutation was reduced to a 2.2-fold difference compared to WT filaments (1.54 × 10-4 ± 0.11 × 10-4 nM-1 s-1).”

Figure 5 B and C: the cosedimentation data do not seem to help elucidate the underlying mechanism. While the authors report statistical significance, differences are small, especially for gel densitometry measurements where the error is high, which suggests that there may be little biological significance. Importantly, example gels from these experiments should be shown, if not the complete set included in the supplement. In B, the higher cofilin concentrations would be expected to stabilize the filaments and thus the curve should be Ushaped.

We do not completely agree with the reviewer on this point. We think the co-sedimentation experiments are useful, as they show that cofilin-1 efficiently binds to mutant filaments, but is less efficient in stimulating disassembly in these endpoint-experiments. This information is not provided by the analysis of the effect of cofilin-1 on the bulk-depolymerization rate and adds to our understanding of the defect of the actin-cofilin interaction for the mutant.

While we agree with the reviewer on the point that co-sedimentation experiments must be repeated several times to produce reliable data, we cannot fully grasp the reasoning behind the statement “While the authors report statistical significance, differences are small, especially for gel densitometry measurements where the error is high, which suggests that there may be little biological significance.”. We interpret this statement as advice to be cautious when extrapolating the observed perturbances of cofilin-mediated actin dynamics in vitro to the in vivo context. We think we are cautious about this throughout the manuscript.

The author expects a U-shape curve, as high cofilin concentrations are reported to stabilize actin filaments by completely decorating the filament before severing-prone boundaries between cofilin-decorated and undecorated regions are generated. We have also performed these experiment with cytoskeletal β-actin and human cofilin-1 and never observed this U shape. This indicates that significant filament disassembly also happens at high cofilin concentrations, most likely directly after mixing of F-actin and cofilin. We cannot rule out that the incubation time plays an important role and that the U-shape only appears after longer incubation times. We also want to direct the reviewer to the publication “A Mechanism for Actin Filament Severing by Malaria Parasite Actin Depolymerizing Factor 1 via a Low Affinity Binding Interface” (Wong et al. 2013, JBC) in which comparable co-sedimentation experiments were performed (Figure 5E-G) with rabbit skeletal α-actin and human cofilin-1 and also no Ushaped curves were observed, even at higher molar excess of cofilin-1 compared to our experiments and with longer incubation times (1 hour vs. 10 minutes).

We now included an exemplary gel showing co-sedimentation experiments performed with WT, mutant actin and different concentrations of cofilin at pH 7.8 in the manuscript (Figure 5 – figure supplement 2)

Figure 5 D: these data show that the binding of cofilin to WT and E334Q actin is approximately the same, with the mutant binding slightly more weakly. It would be clearer if the two plots were normalized to their respective plateaus since the difference in arbitrary units distracts from the conclusion of the figure. If the difference in the plateaus is meaningful, please explain.

As suggested by the reviewer, we normalized the data for a better understanding of the message conveyed.

Figure 6: It is assumed that the authors are trying to show in this figure that cofilin binds both actins approximately the same but does not sever as readily for E334Q actin. The numerous parameters measured do not directly address what the authors are actually trying to show, which presumably is that the rate of severing is lower for E334Q than WT. It is therefore puzzling why no measurement of severing events per second per micron of actin in TIRF is made, which would give a more precise account of the underlying mechanism.

The severing rate as measured in TIRF is dependent on a number of parameters in a nonlinear manner. Therefore, we opted to show the combination of images directly showing the progress of the reaction and graphs summarizing the concomitant changes in cofilin clusters, actin filaments, actin-related fluorescence intensity and cofilin-related fluorescence intensity.

Actin-activated steady-state ATPase data of the NM2A with mutant and WT actin would have been extremely useful and informative. The authors show the ability to make these types of measurements in the paper (NADH assay), and it is surprising that they are not included for assessing the myosin activity. It may be because of limited actin quantities. If this is the case, it should be indicated.

Indeed, the measurement of the steady-state actin-activated ATPase with recombinant cytoskeletal actin is very material-intensive and therefore costly, as a complete titration of actin is required for the generation of meaningful data. Since the vast majority of our assays involving a myosin family member were performed with NM2A-HMM, we decided to perform a full actin titration of the steady-state actin-activated ATPase of NM2A-HMM with WT and mutant filaments. The results of these experiments are now shown in Figure 8C. The panel showing the results used for determining the dissociation rate constants (k-A) for the interaction of NM2C-2R with p.E334Q or WT γ –actin in the absence of nucleotide was moved to the supplement (Figure 8 – figure supplement 2).

We added the following paragraph to the Material and Methods section concerning the Steady-State ATPase assay:

“For measurements of the basal and actin–activated NM2A–HMM ATPase, 0.5 µM MLCKtreated HMM was used. Phalloidin–stabilized WT or mutant F-actin was added over the range of 0–25 µM. The change in absorbance at 340 nm due to oxidation of NADH was recorded in a Multiskan FC Microplate Photometer (Thermo Fisher Scientific, Waltham, MA, USA). The data were fitted to the Michaelis-Menten equation to obtain values for the actin concentration at half-maximal activation of ATP-turnover (Kapp) and for the maximum ATP-turnover at saturated actin concentration (kcat).”

Furthermore, we added a description of the results of the experiments to the Results section of the manuscript:

“Using a NADH-coupled enzymatic assay, we determined the ability of p.E334Q and WT filaments to activate the ATPase of NM2A-HMM over the range of 0-25 µM F-actin (Figure 8C). While we observed no significant difference in Kapp, indicated by the actin concentration at half-maximal activation, in experiments with p.E334Q filaments (2.89 ± 0.49 µM) and WT filaments (3.20 ± 0.74 µM), we observed a 28% slower maximal ATP turnover at saturating actin concentration (kcat) with p.E334Q filaments (0.076 ± 0.005 s-1 vs. 0.097 ± 0.002 s-1).”

(line 310) The authors state that they "noticed increased rapid dissociation and association events for E334Q filaments" in the motility assay. This observation motivates the authors to assess actin affinities of NM2A-HMM. Although differences in rigor and AM.ADP affinities are found between mutant and WT actins, the actin attachment lifetimes (many minutes) are unlikely to be related to the rapid association and dissociation event seen in the motility assay. Rather, this jiggling is more likely to be related to a lower duty ratio of the myosins, which appears to be the conclusion reached for the myosin-V data. These points should be clarified in the text.

We changed the text in accordance with the reviewer’ suggestion. It reads now:Cytoskeletal γ–actin filaments move with an average sliding velocity of 195.3 ± 5.0 nm s–1 on lawns of surface immobilized NM2A–HMM molecules (Figure 8A, B). For NM2A-HMM densities below about 10,000 molecules per μm2, the average sliding speed for cytoskeletal actin filaments drops steeply (Hundt et al, 2016). Filaments formed by p.E334Q actin move 5fold slower, resulting in an observed average sliding velocity of 39.1 ± 3.2 nm/s. Filaments copolymerized from a 1:1 mixture of WT and p.E334Q actin move with an average sliding velocity of 131.2 ± 10 nm s–1 (Figure 8A, B). When equal densities of surface-attached WT and mutant filaments were used, we observed that the number of rapid dissociation and association events increased markedly for p.E334Q filaments (Figure 8 – video supplement 7–9).

Using a NADH-coupled enzymatic assay, we determined the ability of p.E334Q and WT filaments to activate the ATPase of NM2A-HMM over the range of 0-25 µM F-actin (Figure 8C). While we observed no significant difference in Kapp, indicated by the actin concentration at halfmaximal activation, in experiments with p.E334Q filaments (2.89 ± 0.49 µM) and WT filaments (3.20 ± 0.74 µM), we observed a 28% slower maximal ATP turnover at saturating actin concentration (kcat) with p.E334Q filaments (0.076 ± 0.005 s-1 vs. 0.097 ± 0.002 s-1). To investigate the impact of the mutation on actomyosin–affinity using transient–kinetic approaches, we determined the dissociation rate constants using a single–headed NM2A–2R construct (Figure 8D). …..

(line 327) The authors report that the 1/K1 value is unchanged. There are no descriptions of this experiment in the paper. I am assuming the authors measured the ATP-induced dissociation of actomyosin and determined ATP affinity (K1) from this experiment. If this is the case, they should describe the experiment and show the data, provide a second-order rate constate for ATP binding, and report the max rate of dissociation (k2). This is a kinetic experiment done frequently by this group, so the absence of these details is surprising.

In the previous version of the manuscript, the method used to determine 1/K1 (ATP-induced dissociation of the actomyosin complex) was described in the Material and Methods paragraph “Transient kinetic analysis of the actomyosin complex” and the values obtained for 1/K1 were given in Table 1. We now included the experimental data as an additional figure in the manuscript (Figure 8 – figure supplement 3). Furthermore, we also give the maximal dissociation rate k+2 and the apparent second-order rate constant for ATP-binding (K1k+2) for the WT and mutant actomyosin complex in Table 1. Therefore, we changed the paragraph in the Results section concerning this experiment to:

“The apparent ATP–affinity (1/K1), the maximal dissociation rate of NM2A from F-actin in the presence of ATP (k+2), and the apparent second-order rate constant of ATP binding (K1k+2) showed no significant differences for complexes formed between NM2A and WT or p.E334Q filaments (Table 1, Figure 8 – figure supplement 3).”

and the section in the Material and Methods to:

“The apparent ATP–affinity of the actomyosin complex was determined by mixing the apyrase–treated, pyrene–labeled, phalloidin–stabilized actomyosin complex with increasing concentrations of ATP at the stopped–flow system. Fitting an exponential function to the individual transients yields the ATP–dependent dissociation rate of NM2A–2R from F–actin (kobs). The kobs–values were plotted against the corresponding ATP concentrations and a hyperbola was fitted to the data. The fit yields the apparent ATP–affinity (1/K1) of the actomyosin complex and the maximal dissociation rate k+2.

The apparent second–order rate constant for ATP binding (K1k+2) was determined by applying a linear fit to the data obtained at low ATP concentrations (0 – 25 µM).”

For a better understanding of the numerous rate and equilibrium constants, we have now included a figure showing the kinetic reaction scheme of the myosin ATPase cycle (Figure 8 – figure supplement 1).

**Recommendations for the authors:**

**Reviewer #1:**
The subdomains of actin are mislabeled in Fig. 1A.

The labeling of the subdomains has been corrected.

Additional experimental data addressing the 3 weaknesses noted in the public review would be informative but are not essential in my opinion. Examining the effect of cofilin on severing by the TIRF assay in more detail and using a processivity assay for myosin V (immobilized actin) would be the two aspects I would most value.

The TIRF assay for cofilin severing was performed initially over the cofilin concentration range from 20 to 250 nM. The results obtained in the presence of 100 nM cofilin allow a particularly informative depiction of the differences observed with mutant and WT actin. This applies to the image series showing the changes in filament length, cofilin clusters, and filament number as well as to the graphs showing time dependent changes in the number of filaments and total actin fluorescence. We have not included the results for a 50:50 mixture of WT:mutant actin because its attenuating effect is documented in several other experiments in the manuscript.

Our results with Myo5A show a less productive interaction with mutant actin filaments as indicated by a 1.7-fold reduction in the average sliding velocity and an increase in the optimal Myo5A-HMM surface density from 770 to 3100 molecules per µm2. These results indicate a reduction in binding affinity and coupling efficiency, with a likely impact on processivity. Given that Myo5A is only one of many cytoskeletal myosin motors and that the motor properties of all myosins are modulated by the presence of tropomyosin isoforms and other actin binding proteins, we expect only a small incremental gain in knowledge by performing additional experiments with an inverted assay geometry.

**Reviewer #2:**
The authors should address the concerns regarding the statistical methodologies.

We have gone through the manuscript carefully to correct any errors in the statistics, as explained below.

Figure 1B, 5B, 5C, 5D, 8D, 9B, and 8 – figure supplement 2 all show the mean ± SD, as also correctly reported for Figure 8E and 8F in the figure legend. The statement, that these figures show the mean ± SEM was wrong and we corrected this mistake for all the listed figures. Furthermore, we now give the exact N for every experiment in the figure legend.

Figure 2C, 2E, 2F, 4B, 5A, 6B-E indeed showed the mean ± SEM. As the reviewer rightly points out, this is not the appropriate way to deal with such sample sizes. We therefore corrected the figures to show the mean ± SD.

We still refer to the mean ± SEM in Figure 2B, where elongation rates for more than 100 filaments were recorded, and in Figure 8B, where sliding velocities for several thousand actin filaments were measured.

The authors should present the actin titration of the steady state ATPase activity for at least one of the myosins, or preferably all of them.

An actin titration of the steady state ATPase activity of NM-2A has been included in the revised version of the manuscript (Fig 8C).

The authors should consider the use of pyrene-actin in measuring the assembly/disassembly of actin.

Values for the rate of actin assembly/disassembly measured with pyrene-actin are given in Table 1. Based on the small changes observed, we did not determine the critical actin concentration for the mutant construct.